# The SARS-CoV-2 spike protein is vulnerable to moderate electric fields

Claudia R. Arbeitman [1,2,3,5], Pablo Rojas [1,5], Pedro Ojeda-May[4] & Martin E. Garcia [1✉]

Most of the ongoing projects aimed at the development of specific therapies and vaccines against COVID-19 use the SARS-CoV-2 spike (S) protein as the main target. The binding of the spike protein with the ACE2 receptor (ACE2) of the host cell constitutes the first and key step for virus entry. During this process, the receptor binding domain (RBD) of the S protein plays an essential role, since it contains the receptor binding motif (RBM), responsible for the docking to the receptor. So far, mostly biochemical methods are being tested in order to prevent binding of the virus to ACE2. Here we show, with the help of atomistic simulations, that external electric fields of easily achievable and moderate strengths can dramatically destabilise the S protein, inducing long-lasting structural damage. One striking field-induced conformational change occurs at the level of the recognition loop L3 of the RBD where two parallel beta sheets, believed to be responsible for a high affinity to ACE2, undergo a change into an unstructured coil, which exhibits almost no binding possibilities to the ACE2 receptor. We also show that these severe structural changes upon electric-field application also occur in the mutant RBDs corresponding to the variants of concern (VOC) B.1.1.7 (UK), B.1.351 (South Africa) and P.1 (Brazil). Remarkably, while the structural flexibility of S allows the virus to improve its probability of entering the cell, it is also the origin of the surprising vulnerability of S upon application of electric fields of strengths at least two orders of magnitude smaller than those required for damaging most proteins. Our findings suggest the existence of a clean physical method to weaken the SARS-CoV-2 virus without further biochemical processing. Moreover, the effect could be used for infection prevention purposes and also to develop technologies for in-vitro structural manipulation of S. Since the method is largely unspecific, it can be suitable for application to other mutations in S, to other proteins of SARS-CoV-2 and in general to membrane proteins of other virus types.

[1] Theoretical Physics and Center for Interdisciplinary Nanostructure Science and Technology, FB10, Universität Kassel, Kassel, Germany. [2] CONICET Consejo Nacional de Investigaciones Científicas y Técnicas, Buenos Aires, Argentina. [3] GIBIO-Universidad Tecnológica Nacional-Facultad Regional Buenos Aires, Buenos Aires, Argentina. [4] High Performance Computing Center North (HPC2N), Umeå University, Umeå, Sweden. [5] These authors contributed equally: Claudia R. Arbeitman, Pablo Rojas. ✉email: m.garcia@uni-kassel.de

SARS-CoV-2, the agent responsible for the outbreak of COVID-19, is an enveloped virus that utilises its surface glycoprotein spike (S) to bind to the host cell membrane through an angiotensin-converting enzyme (ACE2) receptor[1,2]. Most of the efforts to develop therapies and vaccines against COVID-19[3–9] aim at either decreasing the stability or exploiting some of the structural features of the S protein, since it triggers immune responses and plays an essential role in the virus ability to infect the host[10,11]. The S protein is a homotrimer whose protomers are composed of two functional subunits, S1 and S2, which are responsible for the correct receptor binding/fusion of the viral and cellular membranes, respectively. The S1 subunit contains the receptor binding domain (RBD), which has been found to switch stochastically between a closed ("down") state, in which the receptor binding motif (RBM) is hidden, and an open ("up") state that exposes the RBM thus enabling the interaction and binding with the peptidase domain of ACE2[1,12]. Greater preponderance of the up conformations on mutated S proteins have been linked to higher infectivity, but at the expense of more vulnerability to neutralising antibodies[13]. Distal mutations from the binding region have also been found to affect structural stability of the S protein and its affinity to ACE2, which indicates that a correct spatial arrangement of the RBM residues participating in the binding to the receptor is crucial[14,15]. In addition to this, abundant N-linked glycans decorating the S protein have been found to be involved in both the stability/folding of the protein in its pre-fusion conformation and also in controlling the access of host proteases and antibodies, which provides the virus with support in bypassing the host's immune response[16]. Activation of the membrane fusion is achieved by cleaving the S protein by host proteases at the specific sites located at the boundary between the S1 and S2 subunits. This cleavage triggers a rearrangement from the metastable pre-fusion state into the post-fusion conformation, and the interruption of this process has been shown to prevent viral fusion[17–21]. Altogether, the structure and dynamics of the S protein have been suggested to be the result of a finely tuned balance of affinity to ACE2, stability, and exposure of the RBM[1,2,13,22,23]. Thus, finding new ways to disrupt this balance might result in an additional set of tools to control the virus and therefore the pandemic.

It has been theoretically predicted and experimentally demonstrated that static and time-dependent electric fields (EFs) are capable of inducing conformational changes or even irreversible damage in proteins[24–29]. The fact that extremely intense EFs of strengths larger than 1 Volt per nanometre ($10^9$ V m$^{-1}$) can denature entire proteins and even break chemical bonds is trivial and of little biological relevance. However, the effect of moderate fields is subtler and can be understood in terms of the interactions of the EFs with the permanent dipoles located in the backbone structure and with the additional flexible dipoles on the protein side chains (see Fig. 1d). For instance, under the action of an EF, the electric dipole moments can be reoriented along the field direction in order to minimise the electrostatic energy. On the other hand, a rearrangement of the dipoles can cost conformational energy due to the loss of hydrogen bonds. As a result of the balance between conformational and electrostatic energies along with entropic contributions, the protein can undergo a significant conformational change. So far, studies using molecular dynamics (MD) simulations to address the structural response of proteins to electric fields applied for around 1 microsecond have reported no changes in the secondary structure of proteins and peptides for electric-field intensities below a field strength of $\sim 10^8$ V m$^{-1}$[25,29–32]. In this work, we show, via MD simulations, that EFs of much lower intensities ($10^5 - 10^7$ V m$^{-1}$) cause, on a sub-microsecond time scale, significant damage on the tertiary and secondary structure of the S protein that affects its interaction

with ACE2, potentially making SARS-Cov-2 less infectious. These results pave the way to a range of possible applications of EFs to control structural changes in virions with SARS-CoV-2 being one of the multiple targets.

## Results

**Moderate electric fields induce global long-lasting structural changes in the spike glycoprotein of SARS-CoV-2.** We studied the effect of external EFs on the secondary and tertiary structures of the S protein by performing molecular dynamics simulations. We first considered a representative selected segment of S from a protomer in the conformation "up" between residues 319 and 686. This segment corresponds to a part of the S1 subunit and includes the whole RBD, the subdomains SD1 and SD2, and the interface between S1 and S2 (Fig. 1b). Previous computational[33–35] and experimental[36,37] works on the isolated S protein and also on the related S protein in SARS-CoV-1 have shown that a standalone segment comprising the RBD and its neighbouring subsequence preserves the local structure and therefore the dynamical and biochemical properties that it shows in the entire protein complex. Based on this knowledge we simulated a restricted spatial domain without loss of generality (see also Methods). To construct the initial protein conformation for the simulations, we used the cryo-EM structure PDB ID 6VSB obtained from the Protein Data Bank[1] and completed the missing residues (see Methods). The first production run was aimed at thermalising the system in the absence of EF (no-EF run) in order to bring the protein to thermodynamic equilibrium at 30 °C. An estimate of the free energy profile (see Methods) reveals that motion during the thermalisation run was confined to a single free energy basin (Fig. 2b). This result indicates that the initial conformation fetched from the experimentally obtained segment was close to a stable equilibrium folding state, and the thermalisation run merely helped to relax the remaining structural stress.

Next, using the thermalised structure as initial state, we carried out simulations on the S-protein fragment under the action of an EF during 700 ns. We performed different runs (EF-on runs) corresponding to different EF intensities. The field intensities were selected to span a range between $10^4$ and $10^7$ V m$^{-1}$ in order to cover both low and moderate intensities that are not incompatible with living organisms and can even exist inside cells[38,39]. Only for the sake of comparison, we also performed a short simulation for an unrealistically high intensity ($10^9$ V m$^{-1}$). In all cases, trajectories display an elongation of the protein as a result of the alignment of permanent local dipoles and displacement of charges parallel to the EF (Fig. 2). For the extreme case of EF = $10^9$ V m$^{-1}$, the structural changes are so dramatic that a complete loss of the secondary and tertiary structures of the protein occurs within few ns (see Supplementary Fig. 1). In contrast, for low to moderate intensities (EF < $10^7$ V m$^{-1}$), the field-induced structural changes in the S protein are characterised by a transition to a new stable conformation within a few hundreds of nanoseconds (Fig. 2a, EF-off). For EF = $10^7$ V m$^{-1}$, the protein structure undergoes, besides the above-mentioned transition, an additional structural change in the region between the SD1 and SD2 subdomains, as shown in Fig. 2a and 2d and discussed below. The conformational changes of S upon application of an EF are reflected in the time evolution of the root-mean-square displacements (RMSD) of the protein backbone relative to the starting structures (Fig. 2a). A transition from a conformation exhibiting stable RMSD values below ~0.5 nm to a new stable structure showing small RMSD-oscillations around a larger value occurs within the first 200 ns. For EF = $10^7$ V m$^{-1}$ the unfolding between SD1 and SD2 completes only after 500 ns showing a shift to higher values of RMSD around ~1.9 nm. Taken

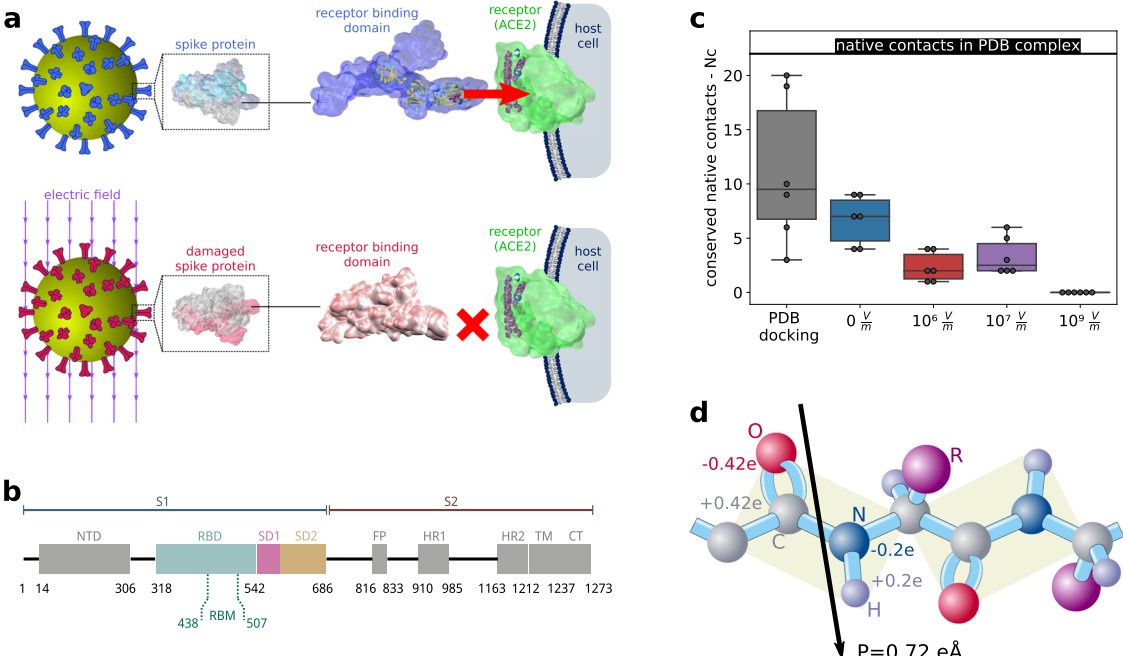

**Fig. 1 External electric fields affect the attachment of SARS-Cov-2 to the host cell. a** Virus entry into the cell is mediated by the recognition between the spike glycoprotein (S protein) present in the virus envelope and the angiotensin-converting-enzyme receptor (ACE2) of the host cell membrane. The binding between the S protein and ACE2 can be altered when external electric fields induce drastic conformational changes and damage in the S protein. **b** Sequence of the S protein (PDB IDs: 6VSB and 6MOJ[1, 19]). Highlighted in colours is the segment used in this study. **c** Conserved number of native contacts ($N_C$) between residues of S and ACE2 for different magnitudes of the EF strength. $N_C$ is maximal for native S protein. Very strong electric fields ($10^9$ V/m) disable the protein by largely deforming its shape, leaving a structure which is unrecognised by ACE2 ($N_C = 0$). Moderate electric fields, which can be induced by available industrial or laboratory devices[71], strongly reduce $N_C$ and are therefore candidates to decrease the affinity of S to ACE2 and, consequently, the infectivity of the virus. The best 6 matches (higher Nc) are taken for each condition for comparison. In the box-whisker plot, the central line indicates median, box limits indicate upper and lower quartiles, and whiskers specify maxima and minima. Source data are provided with this paper. **d** Changes in the structural conformation of proteins under EF are driven by reorientation of electric dipoles.

together, these results suggest that EFs modify the free energy balance enabling the protein to overcome barriers, which in turn results in a shifted conformational ensemble.

To further assess the stability of the new conformations adopted by S under the external fields, we switched off the EF and continued the simulation in absence of fields (EF-off run). Typically, not more than 200 ns were needed for each EF-off run, since for all EF intensities the protein displayed a restricted motion around the structure left after the field application, as revealed by the RMSD plots. For instance, the unfolding of the region between SD1 and SD2 observed under $EF = 10^7$ V m$^{-1}$ remains unaltered after switching off the EF. For all studied EF intensities, estimated free energy plots (see Fig. 2b and Supplementary Fig. 2) confirmed the existence of a new stable minimum and interestingly, they also show that an energy barrier prevents a transition back to the original conformation (Supplementary Fig. 2). Notice that the middle panel of Fig. 2b shows both the initial and the "damaged" states, but both under the influence of the EF. We have also determined the free energy profile connecting both states in absence of fields (see Methods and Supplementary Fig. 3). A barrier separating both states can be clearly observed. Moreover, the figure shows that the minimum after field application is not simply a rapidly decaying metastable state but rather a quite stable, long-lasting state. To visualise the relevant conformational changes, we conducted a principal component analysis (PCA) on the trajectories of the EF-on runs. We considered a subspace spanned by the two most relevant principal components for the run at $EF = 10^7$ V m$^{-1}$ (see Methods). Then, we projected the trajectories for all EF-off runs onto that plane (Fig. 2d). Under the action of EFs of different

intensities the protein goes through different paths in the phase space. The final conformation after each EF-on run depends on the field intensity. After EF switch-off and during the EF-off runs the protein structure remains around the EF-induced new conformations. No return to the initial structures was observed. This can be clearly seen in Fig. 2d, where points corresponding to the trajectories cluster around the final states with almost non-overlapping regions in the reduced phase space (Fig. 2d). The qualitative difference of the states explored by the protein during transient and relaxation driven by static EFs only differing in magnitude suggests the possibility of designing EFs to achieve a predetermined structural change. Altogether, these results provide further evidence that the EF-induced conformational changes in the S protein are long lasting and do not reverse upon removal of EF.

Considering clustered residues as rigid bodies, global conformational changes can be roughly described by the angles formed by the vectors connecting centroids of the domains of interest[40]. We quantified the influence of EF on the unfolding process observed in the region between SD1 and SD2 (see Fig. 1b) by the angle $\theta$ formed between the vectors that connect the centroids of RBD, SD1 and SD2 (Fig. 2d). The difference $\triangle \theta$ between the average values before and after EF application increases monotonously with the EF intensity. The shapes of the distributions during the EF-off runs (Fig. 2e) are clearly clustered with partial overlaps. It is, however, important to point out at this stage, that the angle-shifts for increasing EF do not merely mean quantitatively different versions of the same structural change. Instead, the average value of and its distributions (see Fig. 2e) are determined by different EF-induced tertiary structure

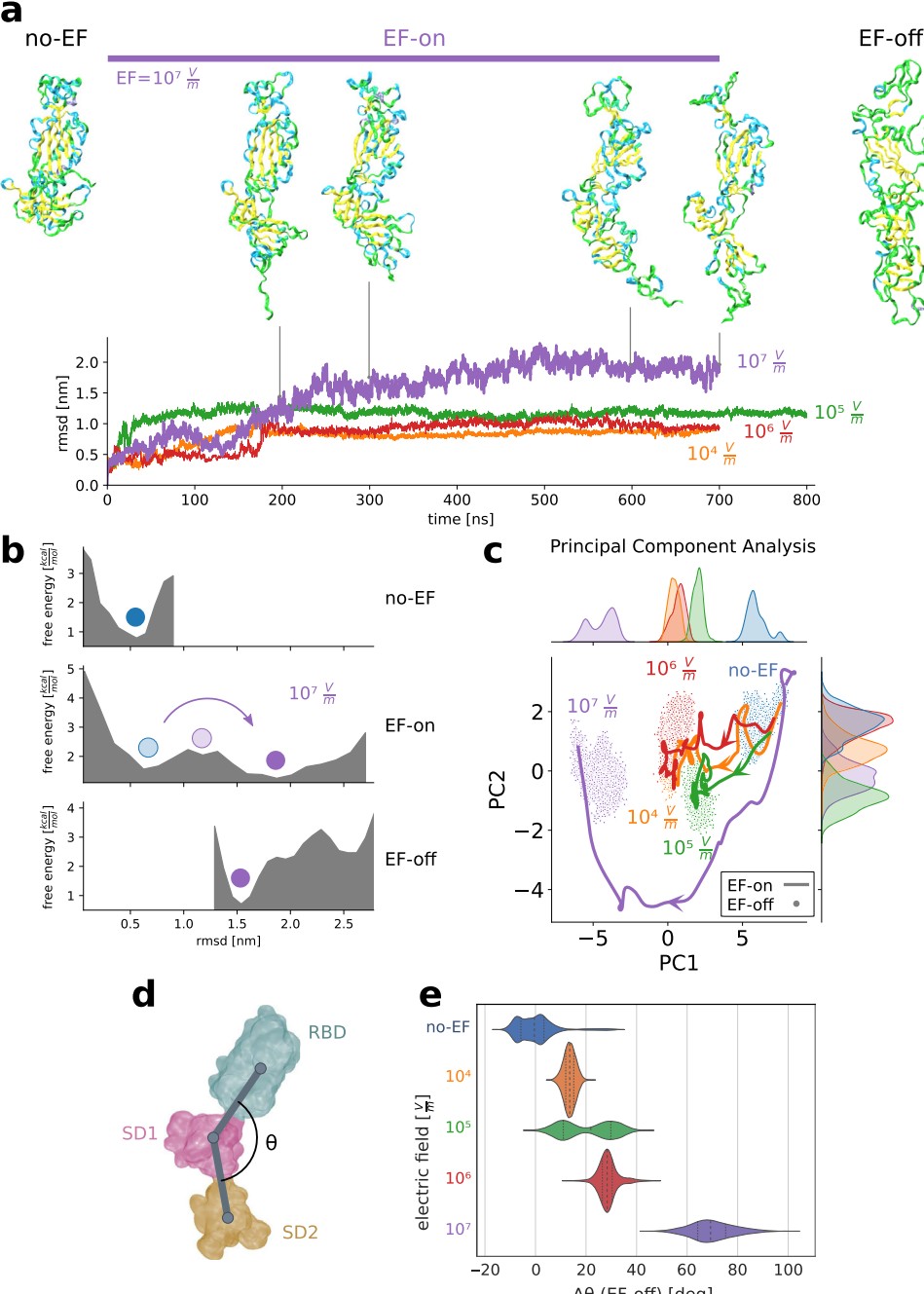

**Fig. 2 Electric fields are able to induce global conformational changes in the spike glycoprotein, affecting the stability of folding states. a**, **b** EF driven major shape changes occur in the different subunits and between subunits of the S protein. **a** Snapshots of the studied fragment of S under an EF of $10^7$ V m$^{-1}$ at 0 ns (initial thermalised stable conformation), 200, 300, 600 and 700 ns, and after EF-off (see text) dynamics during 200 ns. The orientation of the protein is the same in all figures. Trajectories for different electric-field strengths are quantified through the root-mean-square-displacement (RMSD) with respect to the initial structure. Snapshots in **a** correspond to different times along the EF-on trajectory. **b** Electric fields modify the free energy landscape enabling the protein to overcome potential barriers. Estimated free energy landscape along the thermalisation (no-EF-), EF-on- and EF-off trajectories (see Methods). The blue and the light blue dots identify the energy minimum of the initial structure before and during EF application, respectively. Purple dots correspond to the new minimum reached under the EF, which remains stable after switching off the EF. **c** Principal component analysis (PCA) reveals the existence and stable nature of new states after EF application (see Methods). Discretised trajectories of the EF-on and EF-off runs projected onto a plane defined by the two principal components (PC1, 20% of variance; and PC2, 8.2% of variance). Curves on the upper and right axis show the density of points along PC1 and PC2, respectively. Once the S protein has found a new equilibrium basin, which is different for each EF intensity, no return to the initial state occurs after switch-off of the EF. For clarity of representation, curves in the EF-on trajectories are low-pass filtered using a Gaussian kernel (standard deviation 10 ns). **d** Field-induced conformational states can be characterised by the angles formed by the vectors connecting the centroids of clustered residues. **e** Violin plot of the distributions of the shift $\triangle\theta$ of the angle $\theta$ for different field intensities (EF-off runs) with respect to a no-EF representative structure. $\triangle\theta$ is suitable to describe the unfolding of the domain SD2 observed in **a**. In the violin plot, the central line indicates the median, while left and right lines indicate lower and upper quartiles, respectively. EF intensities are color-coded equally for all sub-figures.

rearrangements on other parts of the protein not captured by the coordinate $\triangle\theta$. This means that, for each EF, the accessible angles are constrained by barriers of different origin. This confirms our previous observation that conformational changes for different intensities are qualitatively different in the PCA space and not just amplified versions of the motions at lower intensities (Fig. 2c). Note, that this effect is similar to that induced by mutations. It was, for instance, reported that certain mutations are able to change the statistics of accessed states at distal sites, namely favouring "up" conformations of RBD by mutation of residue 614[37,41]. These results show the potential of EF of different strengths to induce changes in S that do not only affect the overall configuration, but also reshape local interactions.

**Moderate electric fields strongly affect the binding of the spike protein with the ACE2 receptor.** To study how EF specifically disturb the stability of the RBD and, in particular, of residues that are vital to the local interaction with ACE2, we performed MD simulations using a model for the structure of the unbound RBD obtained from a crystallographic experimental structure of the RBD–ACE2 complex (PDB ID 6M0J, chain E)[19]. We first conducted thermalisation simulations under EF = 0 V m$^{-1}$ (no-EF), which showed that the tertiary structure is preserved compared to the initial crystal structure (see Fig. 3). Previous studies[42,43] have shown that the RBD can be described as being formed by a core and the RBM. The loop 3 (L3), between residues Tyr470 and Pro491, is one of the four loops comprising the RBM, and has been demonstrated to play a key role in the interaction of S with ACE2[19]. The presence of two small $\beta$-strands in the fragments Cys488–Tyr489 and Tyr473–Gln474 of L3 was shown to be one of the reasons for the enhanced affinity of SARS-CoV-2 with ACE2, which is 15–20 times larger than the affinity of SARS-CoV-1, whose S protein exhibits a L3 loop without sheets[1,40]. The larger affinity to ACE2 makes SARS-CoV-2 much more infectious than SARS-CoV-1[15,44]. The above-mentioned beta-strands in L3 remained intact along the thermalisation simulations, in agreement with previous studies showing that the greater rigidity of the $\beta$-sheets increases the stability of L3 in SARS-CoV-2 as compared to the unstructured L3 in SARS-CoV-1[15,44–48]. Furthermore, the rest of the secondary structure of the RBD was observed to be stable during the thermalisation run (Fig. 3a, left structure). These results are consistent with current studies providing evidence for the stability of the secondary structure of the RBD, and set a baseline for a comparison with the RBD structures affected by EFs.

We next performed EF-on simulations for different field intensities, namely EF = $10^5$, $10^6$, $10^7$, $10^9$ V m$^{-1}$, followed by the corresponding EF-off simulations. During the EF-on simulations, the secondary structure of the RBD was disrupted at multiple segments. Particularly, L3 undergoes a transition from the close structure with the two beta-sheets to an open and completely unstructured coil, reminiscent of L3 in SARS-CoV-1[15,49] (Fig. 3a, right structure for EF = $10^7$ V m$^{-1}$). We evaluated the time evolution of the secondary structure of L3 (see Fig. 3b for E = $10^6$ V m$^{-1}$ as example), which shows that beta sheets gradually shrink during the first 200 ns of the simulation, until they finally deconstruct as turns or random coils before 1 $\mu$s. This indicates that the stretching of the protein by an external electric field leads to a destabilisation of the initial conformation. In the subsequent EF-off simulation, beta sheets do not recover and L3 remains stable in its open unstructured state (Fig. 3b). Coil or loop structures in proteins were previously described to have higher flexibility than highly ordered secondary structures such as $\beta$-sheets and helices[50]. We quantified the changes in flexibility of RBD by computing the root-mean-square fluctuations (RMSF) of the RBD (Fig. 3c). RMSF plots reveal that the EF modifies the

flexibility of RBD inhomogeneously, with particular emphasis in the L3 loop and the RBM, in general. These results provide evidence that application of EF changes the secondary structure enduringly in segments that are critical for the interaction of the RBD with ACE2, and disrupts the spatial atomic organisation of the backbone and side chain in key residues.

We also focused on the effect of EF in the spatial distribution of key residues of the RBM. We particularly analysed the residues that were previously described as participating in stabilisation of the RBM and in the interaction of the RBD with ACE2, since they contribute to the formation of a network of hydrogen bonds, hydrophobic and electrostatic interactions[51,52]. For instance, pairwise interactions between residues Cys488-Gly485 and Gln474-Gly476 have been pointed out as responsible for the stabilisation of L3[15]. In the no-EF simulation, the corresponding distances between residues were observed to remain within values around 4–5 Å that enable those interactions[53] (Supplementary Fig. 3). During EF-on, the same distances increase up to values between 8 and 10 Å, which causes a weakening of the inter-residue interaction. Another important set of residues of the RBM involved in hydrophobic contacts with the central region of the N-terminal helix of ACE2 are localised at the L2 and L3 loops, mainly comprising the aligned amino acids Leu 455, Phe456, Tyr473 and Tyr 489. In the original crystal structure 6M0J and in no-EF simulations, the aromatic residue from Phe456 is in close contact (less than 6 Å) with Tyr473 and the amino group of Lys 417, which leads to a very important stabilising internal $\pi$-cation interaction[52]. Application of an EF causes the following structural reorganisation of this set of residues: the increased mobility of L3 leads to a break of the Phe456 close-contact interactions, which reorders and misaligns this sequence of hydrophobic contacts (Supplementary Figs. 4 and 5). To complete the analysis of the EF-induced damage on the RBD, it is important to consider the residue Phe486, which plays a major role in the interaction of S with ACE2 because of its penetration into a deep hydrophobic pocket of ACE2[41]. This interaction consists of a $\pi$-stacking of Phe486 with Tyr83 and two intermolecular contacts with the side chains of Leu79 and Met82 of ACE2, which contribute to L3 stability and the enhanced receptor binding. Figure 3d shows the RBM from the PDB structure 6M0J, from a representative conformation at EF = 0 and after the EF-off run for EF = $10^6$ V m$^{-1}$. Notice that Phe486 is exposed in the $\beta$-turn of L3 in the absence of field, while the structure reorganisation by EF hides this residue in a persistent $\beta$-coil making it sterically inaccessible for the hydrophobic patch of ACE2. The above presented findings suggest that an EF induces disorder at sub-nanometre level that leads to unfavourable positions and orientations of important residues involved in the stabilisation of the RBM and the interaction with ACE2.

To further confirm that the atomic reorganisation in the S protein caused by the EF is likely to weaken its interaction with ACE2, we computed the electrostatic potential $\phi$ of the RBM for the crystalline structure and for the final structures of the no-EF and the EF-on runs, respectively. Electrostatic interactions have been intensively studied due to their importance in biomolecules recognition and binding[54,55]. We computed $\phi$ by solving the Poisson-Boltzmann equations for continuum electrostatics using the APBS package[56,57]. Figure 3e reveals, exemplifying for EF = $10^7$ V m$^{-1}$, that the spatial distribution of $\phi$ on the RBM is severely distorted upon EF application. In particular, the surface charge distribution in the L3 region is strongly affected. Notice that in the crystal structure (PDB ID 6MJ0) the binding surface on the ACE2 side exhibits a positive patch in the central region (blue area, contributed by residue Lys 31) that matches with the corresponding negative area on the RBD, bounded by a set of polar and acidic residues including Glu471, Thr478, Glu484 and Gln493[44]. These matching areas contribute to the strong electrostatic

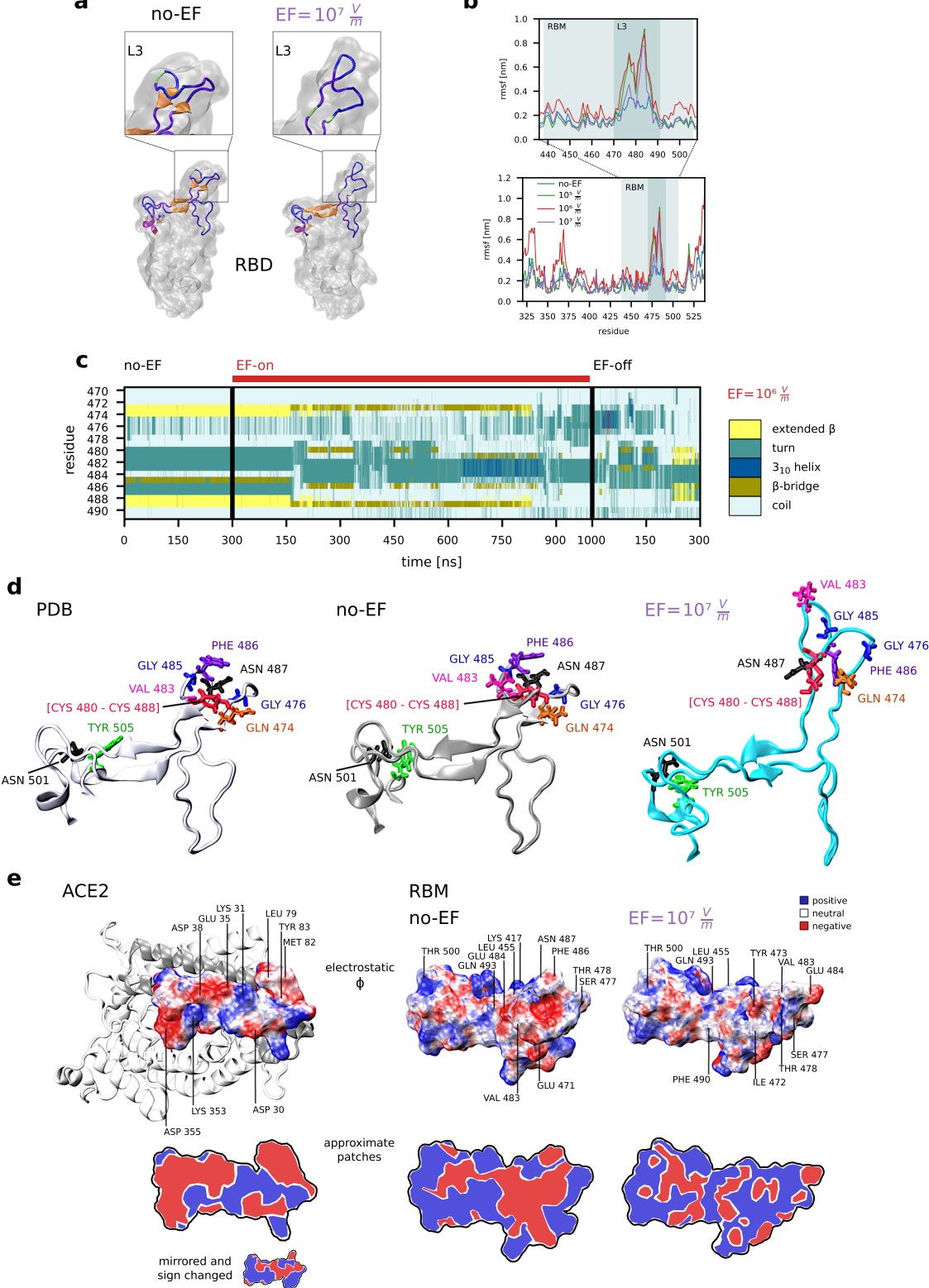

complementarity at the binding interface. After thermalisation (no-EF run) no significant changes occur in this region. However, upon rearrangement of L3, and especially the charged residues Glu471 and Glu484, due to EF application (EF-off run), the negative region in the RBD shifts and faces the negative part of ACE2, generating a repulsive force. At the same time, the region of the RBD opposite to Lys 31 of ACE2 exposes non-polar residues. These calculated electrostatic properties indicate that surface charge distribution in

the RBM is strongly modified by EF and therefore the electrostatic complementarity between the S protein and ACE2, and therefore bonding between the RBD and ACE2 is disrupted (see Fig. 3e as example for $EF = 10^7 \, V \, m^{-1}$). Docking tests (see below) verified that these contacts are lost.

Finally, and in order to estimate the impact of positional and orientation changes of the residues along with charge and dipole rearrangements on the binding of RBD with ACE2, we performed

**Fig. 3 The secondary structure of the RBM can also be irreversibly perturbed by electric fields, affecting residues that participate in the binding to ACE2. a** The recognition loop L3 (Tyr470 to Pro491), exhibits two parallel $\beta$ sheets, which are responsible for a higher affinity to ACE2[15, 42]. The electric field induces a change of the secondary structure of L3 to an unstructured loop (example for EF $= 10^7$ V m$^{-1}$). The spatial arrangement of key recognition residues is completely altered, which most likely inhibits their recognition function. **b** Root-mean-square fluctuations of the amino acids of the RBD for different EF strengths. Residues in the receptor binding motif are highlighted and zoomed-in, showing the increased flexibility after EF application. **c** Temporal evolution of the secondary structure of the L3 loop of the RBM (residues 470–491) for the no-EF, EF-on and EF-off runs (EF $= 10^6$ V m$^{-1}$). The secondary structure is lost under field application and does not recover after EF switch-off. **d** Close-up view highlighting the key residues of RBD participating in the binding with ACE2 for the crystal structure 6M0J (left), for a representative snapshot of the thermalised structure at 30 degrees Celsius (centre) and for a representative snapshot of the EF-off run (example for EF $= 10^7$ V m$^{-1}$, right). **e** Electrostatic potential at the surfaces of the relevant docking regions of ACE2 (left) and of the Receptor Binding Motif (RBM) at EF $= 0$ (centre) and EF $= 10^7$ V m$^{-1}$ (right). Red, white and blue potential surface colours indicate negative, close to neutral and positive charges, respectively. The lower panel shows a simplified planar visualisation of the positive and negative patches on both sides of the RBD–ACE2 bonding interface. The small inset shows a mirrored and sign-changed version of the charged surface of ACE2 in order to facilitate the appreciation of the charge complementarity between RBD and ACE2 in the absence of fields and its disruption upon application of an EF.

calculations of the docking between the EF-induced structures of the RBD and ACE2 using the tool PyDOCK[58]. We first listed the native-like contacts between RBD and ACE2[15,44,51] in the 6M0J crystal structure (using a cut-off of 7 Å, see Methods and table in Supplementary Fig. 6). Then, we computed the remaining contacts from the above list in the best docked structures after EF and in absence of EF (including the final no-EF and the 6M0J structures). We selected the cases with the highest number of preserved contacts (see Methods and Supplementary Fig. 6), which are shown in Fig. 1c. The number of preserved contacts in the selected structures are taken as an estimate of how well the resulting structure of RBD can dock to the residues of ACE2, and therefore as a guess of how likely the RBD of S can bind to ACE2. The number of preserved contacts significantly decreases for increasing EF intensity (Fig. 1c) up to the limit where all native contacts disappear for extremely high intensities. These estimates of docking efficiency further support our findings that EFs induce conformational changes disturbing the interaction of the RBD with ACE2 by spatially reorganising amino-acid dipoles and charges.

**RBD mutations corresponding to the variants of concern B.1.1.7 (UK), B.1.351 (South Africa) and P.1 (Brazil) are effectively damaged by moderate electric fields.** The recently emerged mutations of the S protein in the new "variants of concern" (VOC) 501Y.V3/P.1 (Brazil), 501.V2/B.1.351 (South Africa) and 501Y.V1/B.1.1.7 (UK) generate great worries because they exhibit a significantly higher infection rate and circulate globally[59–62]. Important consequences of the circulation of the emerging variants are increased transmissibility, pathogenicity and ability to escape from neutralising antibodies or vaccine-induced response[63,64]. The multiple mutations in the VOC also affect the RBD through critical residues as N501Y, E484K, K417N and K417T[65] which are believed to enhance the interactions between the spike protein of SARS-CoV-2 and the ACE2 receptor[66,67]. In this work, we have also studied the effect of EFs on the RBD of the VOC. As in the case of the wild type, we considered as a starting structure of the RBD the chain E from the PDB structure ID: 6M0J (see Methods), which we used as a template. We then generated the three VOC by the corresponding amino-acid exchanges. We performed different runs (thermalisation no-EF 500 ns, EF-on $10^5$ V m$^{-1}$ 1000 ns and EF-off 300 ns) following the previously applied protocol (see Methods). Remarkably, the secondary structure of the L3 loop of the three considered VOC also undergoes a transition from the closed structure with the two beta-sheets to an open unstructured coil. Subsequent EF-off simulations for 300 ns revealed no differences in the unstructured nature of L3. In Fig. 4 we show the corresponding RBD structures and RBM-electrostatic potentials in absence, under the presence, and after switch-off of an external

field with a strength as low as $10^5$ V m$^{-1}$. This shows that the external electric field has a significant impact on RBD of the mutants which leads to conformational changes in the region localised near the ACE2 interaction interface. These changes are, at least, as serious as for the wild type (see Fig. 4). The structural implications of the VOC mutations on the electrostatic potential at the interface between the RBD of the spike protein and ACE2 for the final structures (after no-EF, EF-on and EF-off runs) were evaluated by solving the Poisson-Boltzmann equations[56,57] (APBS package). As shown in Fig. 4, the spatial distribution of the electrostatic potential all over the RBM is strongly affected in all three variants upon application of an EF. The effect of the E484K mutation in the B.1.351 and P.1 variants manifest itself in a large-amplitude motion and expulsion of Lys484 from the central hydrophilic patch at the interface to ACE2, altering the electrostatic binding. Note, that this movement causes strong rearrangement of the residues and, consequently, changes in side-chain conformation, which should also alter the contact numbers with ACE2. Summarising, our simulations on the VOC predict that for all three considered mutants, an external EF as low as $10^5$ V m$^{-1}$ produces a severe structural damage and conformational reorganisation of the RBD–ACE2 interface.

## Discussion

The structure of the S protein of SARS-CoV-2 seems to exhibit a finely tuned combination of geometrical, physical and chemical properties which provide efficiency in infectivity and bypassing the host immune system[13], as it is well documented in multiple studies on the effect of simple mutations, ligands, antibodies and recombinant protein expression systems[2,5,22,37,41,68,69].

In this work we showed that the application of relatively low to moderate static EFs can persistently change both the secondary and tertiary structures of S by rearranging and reorienting residues, thus disordering originally ordered segments through breaking and rebuilding of hydrogen bonds and salt-bridges. This results in reshaped local interactions and relative displacements between domains. Disruption of the secondary structure of S, particularly at the RBM, occurs under notoriously weaker EFs than those needed to produce significant changes in other proteins[25,29–32]. This suggests a possible causal link between structural vulnerability and affinity to ACE2. The structural features of S allowing the virus to develop its function and to avoid the immune response are in turn those ones particularly unprotected to EFs. The pre-fusion state of the S protein of SARS-CoV-2, like that of other class I viral fusion proteins, is metastable[70]. This seems to be important for optimizing or regulating their functions. Thus, function of viral fusion proteins depends on their ability to fold into a less stable but functionally relevant pre-fusion conformation, and to limit the kinetic accessibility of the

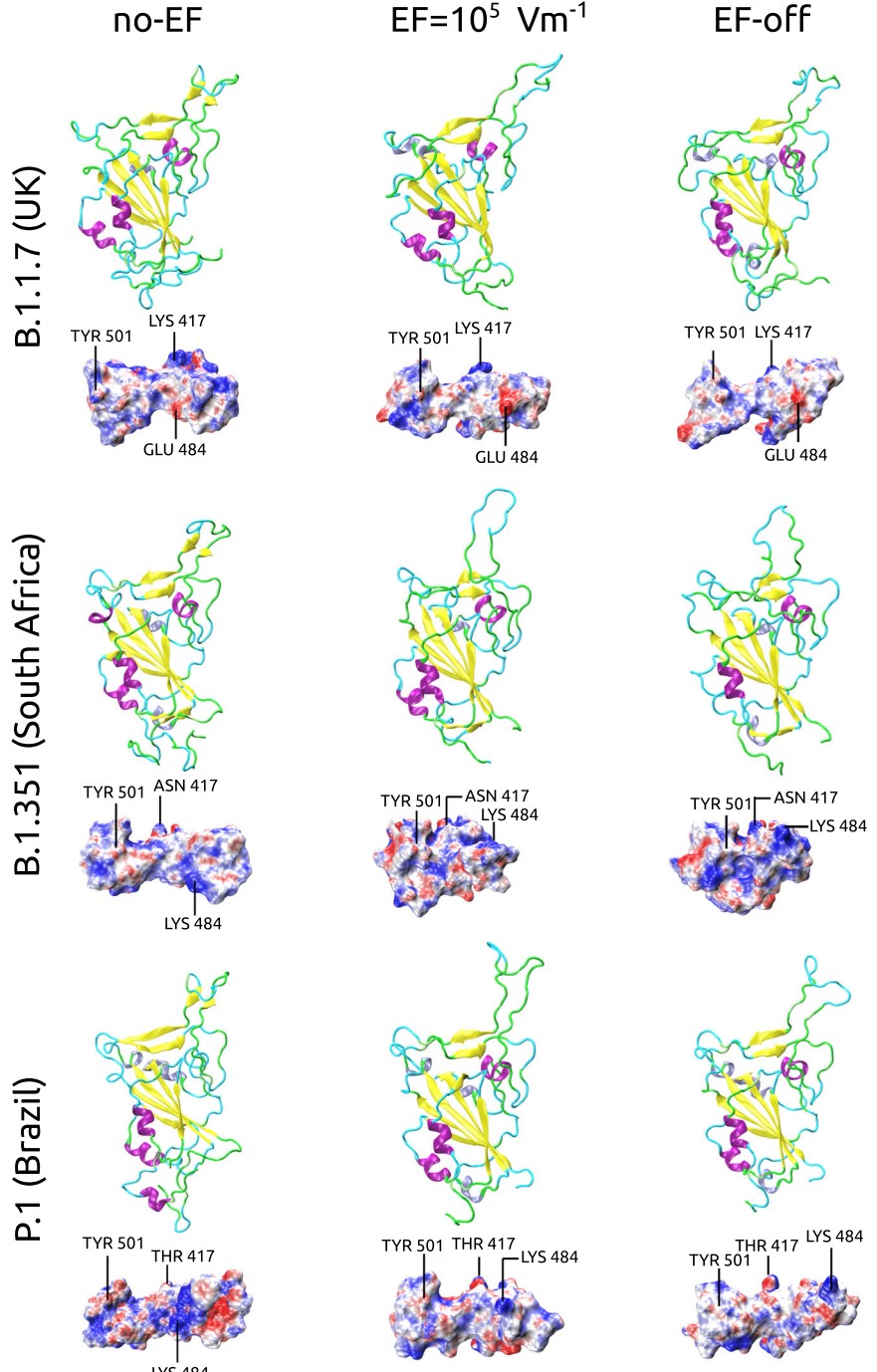

**Fig. 4 Mutants of the RBD corresponding to the variants of concern B.1.1.7 (UK), B.1.351 (South Africa) and P.1 (Brazil) are also irreversibly damaged by electric fields.** Variants were generated by replacing individual residues in silico in the 6M0J structure. The introduced mutations were N501Y (P.1, B.1.1.7 and B.1.351) K417N (B.1.351), K417T (P.1) and E484K (B.1.351 and P.1). For each of the variants, the secondary structures and the electrostatic potential surfaces are shown for the conditions no-EF, EF-on ($10^5$ V m$^{-1}$) and EF-off, in the left, middle and right columns, respectively (details in Methods and caption of Fig. 3). The position of mutated residues is shown on the electrostatic potential surfaces. Upper panel, results for variant B.1.1.7 (UK). Middle panel, results for variant B.1.351 (South Africa). Lower panel, results for variant P.1 (Brazil). In all variants, the secondary structure is severely disrupted by the electric field at the level of the two parallel $\beta$ sheets in the loop L3, that turns to an unstructured coil. The electrostatic potential surfaces change under the EF to a distorted pattern in analogous way as in the wild type (see Fig. 3).

more stable post-fusion conformation. Now, the energy barrier trapping the pre-fusion state was found to be surprisingly low in the case of coronaviruses S proteins[70]. We hypothesise that low to moderate electric fields are enough to modify the energy landscape around the local pre-fusion minimum, inducing a non-thermal transition to a state between the pre- and the post-fusion

conformations. This argument is also supported by our calculation of the free energy profile shown in the Supplementary Fig. 3. As functionally active metastable states are encountered in other viral fusion proteins (e.g. influenza virus hemagglutinin, gp120-gp41 HIV or Ebola virus GP), we expect the unusual vulnerability to EF to occur for those cases.

It is important to stress that EF of most of the intensities described here are achievable in practice in diverse contexts. For instance, EF strengths between $10^6$ and $10^7$ V m$^{-1}$, being well below the dielectric-breakdown threshold of water[71], are commonly used in industrial food processing to inactivate pathogens[72]. The immediate availability of cheap ways to produce EFs along with the basic skills required to manipulate them in both lab and industrial environments indicate that the effects described in this paper could spark the development of multiple solutions to mitigate the COVID-19 pandemic. For example, exposure of infectious microdroplets (aerosols)[73] or samples to EFs for timescales greater than a microsecond can generate inactivated or attenuated virions, and thereby cancel a mode of transmission that has been pointed out as the dominant for COVID-19[73].

Our findings might also open up ways for EF based in-vivo (in the case of low EF strengths such as $10^4$ V m$^{-1}$) and in-vitro therapeutic approaches. Recent works on exposure of mice to EF near to $10^4$ V m$^{-1}$ for several weeks as a treatment for diabetes reported no adverse side effects[74]. In the previous Section we showed that EFs disturb both the shape and charge complementarity to ACE2 and cause the exposure/hiding of key RBM residues, leading to a dramatic reduction or suppression of the docking possibilities of S to the host cell. Owing to the same principle, the EF-induced motion of residues could be exploited to either favour or prevent the interaction of S with other molecules. For instance, cryptic epitopes (sequences that are inaccessible in the pre-fusion state) could be exposed under EFs, allowing for the binding of antibodies or ligands, as an alternative method to mutation-based techniques available in recombinant protein technology[75,76]. For instance, ligands acting as blockers can attach to regions of S and lock RBD up-down transitions[76]. Application of EF under the presence of blockers could yield to both disruption of functional motifs and locking dysfunctional conformations. The complete and ultrafast protein denaturation of the studied segment of S obtained under extremely high strengths ($10^9$ V m$^{-1}$, see Supplementary Fig. 1) after less than 1 nanosecond suggests that intense EF pulses can cause analogous effects as antibodies that inactivate SARS-CoV-2 by premature conversion of S from pre-fusion to post-fusion[20]. Although denaturation of S can also be achieved by raising temperature or changing the pH, the process is not controllable, in contrast to the non-thermal barrier-crossing occurring under EFs[77]. In Supplementary Fig. 8 we also show that the EF produces structural damage in the RBD even when it is bound to the ACE2 receptor.

It is important to point out that our study shows consistent results between simulations based on different PDB structures of S (6M0J and 6VSB). The driving forces governing atomic motion and residue rearrangement depend on dipole-alignment and charge distributions. Therefore, while single mutations can change the structural aspects of S, the driving forces under EFs at the local scale will be of the same order of magnitude throughout most of the sequence. We have shown in this paper that the mutated types Variants of Concern B.1.1.7, B.1.351 and P.1 are at least as strongly damaged by EFs of low to moderate strengths as the wild-type RBD. This occurs because their multiple S-protein mutations involve electric charge changes (from neutral to negative in A570D, from neutral to positive in P681H, from negative to positive in E484K and D1118H), while the native pre-fusion conformation of S is mostly preserved. Other mutants not analysed here, such as the VOC B.1.617 (India), also involve charge substitutions that have been suggested to enhance electrostatic interaction of the RBD with the ACE2 receptor[78]. The concrete case of the sub-variants B.1.617.1 (L452R changes from neutral to positive and E484Q from negative to neutral), B.1.617.2 (L452R changes from neutral to positive and T478K from neutral

to positive) and B.1.617.3 (L454R changes from neutral to positive and E484Q from negative to neutral)[79,80] clearly reflect this fact. Therefore, we expect that mutants labelled as variants of interest, under monitoring or that may arise in future, will also be particularly vulnerable to EF. The same should hold for the spike proteins of other viruses.

Last but not least, and on the same line, the application of EFs to SARS-CoV-2, which strongly reduces its infectivity through modification of S, might also offer the possibility to generate, though applications in air filters or masks, partial immunity or cross reactivity against wild-type SARS-CoV-2 and its different mutations. Since there are some evidences that immunological memory due to infection with seasonal human coronaviruses (hCoVs) may generate cross-protection to SARS-CoV-2[81–85], the EF-modified long-lasting states of the S proteins of SARS-CoV-2 might still provoke a certain immune response to the wild-type virus.

Summarising, this study demonstrates that EFs of different biologically relevant strengths change S of SARS-CoV-2 both at nanometre and sub-nanometre scales. Considerable changes in the secondary structure of the RBD in the wild type and currently dominant mutants of S occur at field strength orders of magnitude smaller than for most proteins[24–32]. We conclude that the spike protein of SARS-CoV-2 (and especially its RBD) is unusually vulnerable to external electric fields. Results of Fig. 2 show that the ensuing states under EF application clearly represent distinct atomic rearrangements depending on field strength. This raises the question whether tailored EF could be designed in order to drive S towards desired target structural states. Pulse trains, like those used in the food industry, or shaped oscillatory EFs of variable central frequency, envelope, duration and polarisation, could be optimised to promote a selective structural response in a similar way as in concepts involving electromagnetic fields[86].

## Methods

**Protein structures preparation.** The initial conformation (including atomic coordinates) was obtained from two available PDB structures with IDs: 6VSB and 6M0J, respectively. We considered in our simulations part of the chain A (residues 319–686) from 6VSB and the chain E (residues 333–526) from 6M0J, corresponding to the S protein and RBD, respectively. The missing hydrogen atoms and residues were added by using the CHARMM software (v. 45b1)[87] and CHARMM-GUI[88] with the CHARMM-36 force field parameters[89–91] for A, and the tool Modeller[92] for E. Residues were protonated to fulfil pH 7 conditions and histidine (His) residues were treated as protonated on ND1 state (HSD). N-acetyl-β-glucosaminide (NAG) glycans were kept as in the original crystal structures and were modelled with the parameters of 2-acetyl-2-deoxy-β-D-glucosamine, with CONH fully charged atoms, using the same CHARMM-36 force field.

**Molecular dynamics simulations.** The simulations were performed using the GROMACS package (version 2019.4)[93–95]. CHARMM-36 force field parameters were adopted[89]. Both systems, 6VSB and 6M0J, were solvated with 298746 and 154770 TIP3P water molecules[96], respectively, with periodic boundary conditions. Na$^+$ and Cl$^-$ ions were further added to the boxes to simulate a salt concentration of 150 mM. The total number of atoms of segments A and E were 304841 and 158681, respectively. The LINCS algorithm was used to constrain all hydrogen bonds[97,98]. A cut-off of 12 Å was used for both the van der Waals and electrostatic interactions. The latter were computed with the help of the PME method[99,100] using a fourth order of cubic interpolation scheme with a grid size of 1.2 Å.

First, the energy of the system was minimised, while keeping heavy atoms at initial positions with harmonic constraints on backbone and side-chain atoms of 95.6 and 9.56 kcal/mol nm. Further restraints were applied on dihedral angles on atoms 3420, 3422, 3424, 3425, 3435, and 3439 with a force constant of 0.22 kJ/mol/rad$^2$ until the forces were less than 239 kcal/mol nm. This step was followed by a short equilibration stage of 1 ns using the NVT ensemble with the Nose-Hoover thermostat and a time constant coupling of 1 ps. Then, a longer equilibration run of 100 ns using the NPT ensemble was performed, in which the barostat was simulated using the isotropic Parrinello-Rahman algorithm[101] with a time constant coupling of 5 ps, a compressibility of $4.51 \times 10^{-5}$ bar$^{-1}$ and a reference pressure of 1 bar. The parameters of the thermostat used in both ensembles are the same.

The production runs were performed in the NPT ensemble. Once an equilibrated trajectory with no-EF was obtained, we used the atomic coordinates at 100 ns (PDB 6VSB) and 300 ns (PDB 6M0J) as the reference structures. In order to

speed up calculations, EF-off runs based on PDB 6VSB were started using atomic coordinates of the EF-on runs at 500 ns, and EF-on were continued until a minimum of 700 ns to ensure that no further major atomic displacements occurred in the remaining simulation time. EF-off runs using PDB 6M0J were started using the final atomic positions from the EF-on runs after 1 μs.

**Application of the electric field**. The electric fields were applied in x-direction with respect to the MD simulation box. The electric-field couples to all charges in the system, including the charged atoms in the protein, in the water molecules and the isolated Na$^+$ and Cl$^-$ ions. The interaction of the system with the external static electric field $E = (E_0, 0, 0)$ is introduced by an additional force term of the form $F_i = Eq_i$ acting on atom $i$ in the MD cell, where $q_i$ refers to the atomic charge. The protein is allowed to freely rotate. During the production runs the protein rotated around many different axes, as was confirmed by visual inspection of the results. This means that there is no preferred direction for the application of the electric field on the protein. Note that this resembles a real situation in which the virus rotates in the space between two electrodes where the field is acting.

The EF strengths mentioned throughout this work refer to the applied external field. Since we consider the coupling of all charged atoms, including ions in the solution and the H$^-$ and O$^-$ atoms of water, with the EF, the water molecules become polarized, generating an electric field opposite to the applied one, effectively screening it. Therefore, the protein is indeed affected by a total field composed by the vectorial sum of the external electric field and the induced field due to the water polarisation[102].

**Validation of the force field**. For the sake of validation of the non-polarisable CHARMM-36 force field, simulations with the Drude polarisable force field[103–105] were performed using the CHARMM-GUI and CHARMM (v 45b1) software. The initial structure was the same as for the simulations using the CHARMM-36 force field except for the glycosylated residues which were modelled without sugar rings. The cubic box had an initial volume of $(100 Å)^3$ and a final size of $(97.8 Å)^3$ after equilibration. These simulation sets were run with the NAMD software (v. 2.14). The resulting system consisted of 156,541 atoms including 86 Na$^+$ and 93 Cl$^-$ ions. Electrostatic interactions were solved with the Particle Mesh Ewald method with a grid spacing of 1.5 and splines 6$^{th}$ interpolation order. A Drude temperature of 1 K was used, with a damping coefficient of 20 ps$^{-1}$, a bond length of 0.2 Å, and a Drude force constant of 40,000 kcal/mol Å$^2$. Also, a non-bonded Thole interaction radius of 5 Å together with a Drude hard wall option were considered. Initially, 10000 minimisation steps were done to avoid atomic clashes followed by a 1 ns equilibration run by using the Langevin thermostat with a damping coefficient of 5 ps$^{-1}$ including the hydrogen atoms. Pressure was controlled using the Nose-Hoover Langevin barostat with a target pressure of 1.01 bar, piston period of 50 fs and piston decay of 25 fs. The time-step and temperature were 0.1 fs and 303.15 K, respectively. The production runs under EF application were 0.2 ns long.

Supplementary Figure 10 shows the comparison of the results for the root-mean-square fluctuations inside the RBM for different EF strengths using both force fields. The similar results, particularly concerning the magnitude of the fluctuations, confirm that the force field CHARMM-36 is accurate enough for this kind of simulation. Notice that long-time simulation using the polarisable force field would be computationally not affordable within a reasonable computer time due to the ultrashort time-step needed (100 attoseconds). Moreover, polarisable force fields exhibit instabilities.

**Analysis of simulations**. Unless otherwise stated, trajectory files were read and post-processed using GROMACS tools or the MDAnalysis Python library[106]. The VMD software[107] was used to visualise the MD trajectories and to draw the molecular representations.

*Principal components analysis (PCA) of the EF-on and EF-off trajectories.* In order to represent and visualise trajectories and states, we projected each of them in a suitable two-dimensional space obtained by dimensionality reduction. First, we computed and used dihedral angles as the generalised coordinates defining structural states, instead of the atomic cartesian coordinates. Dihedral angles, due to the local nature of its definition, are suitable to naturally separate internal motion from the overall motion of the protein. Furthermore, we transformed dihedral angles by splitting each one into two metric coordinates corresponding to its sine and cosine components[108]. This transformation from dihedral space to a linear metric space with a well-defined Euclidean distance, has been shown to preserve a unique representation while avoiding artifacts arising from the periodicity of angles[109]. Next, we found the reduced space by performing PCA over the above-mentioned metric coordinates and selected the first two components (corresponding to the highest eigenvalues) as the representative directions defining the reduced two-dimensional space. PCA finds the directions of correlated motion by diagonalising the covariance matrix, with eigenvectors representing the directions of collective motion and eigenvalues ranked in descending order representing their amplitudes. We performed PCA using the scikit-learn Python library[110]. Then, we projected each trajectory onto the reduced PCA space. Although we used the components corresponding the trajectory under EF $= 10^7$ V m$^{-1}$ to project all results, similar

plots arise if principal components corresponding to runs of other EF intensity are used. Therefore, the conclusions are independent of this choice.

Different residues and atomic distances (e.g. for disulfide bond and key amino acids) were calculated using customised Python scripts and the MDAnalysis library. Centroids of subdomains (for angle calculations) were computed as the arithmetic mean of the positions of the set of α-carbons of the corresponding residues. Using α-carbons speeds up calculations and yields negligible differences in the position of centroids with respect to the case where all atoms are considered. For determining the distance between individual residues to analyse interactions, we computed the mean distance between all the atoms of each residue.

The changes on the secondary structure of the RBD over time with no-EF, EF-on and EF-off were estimated by the STRIDE algorithm implemented in the VMD software package version 1.9.4a38 (2019). The stride algorithm relies on hydrogen bond energy together with statistically derived backbone torsion angle data for the secondary structure characterisation in trajectories previously obtained by GROMACS.

*Free energy profile estimate.* The free energy was estimated along the RMSD as a reaction coordinate using a path-sampling method[111–113] to approximate the potential of mean force (PMF) for each condition (no-EF, EF-on and EF-off) and for each EF strength. The free energy profile is then estimated by

$$F(rmsd) = -k_B T \ln(\langle \delta(rmsd_k - rmsd) \rangle) \qquad (1)$$

where $rmsd_k$ is the windowed RMSD value of the $k$ position along the path, $k_B$ is the Boltzmann constant, $T$ is the temperature and $\delta(\dots)$ is the Dirac delta function. Each path was binned using 20 windows along the RMSD coordinate.

The free energy profile in Supplementary Fig. 3 was generated as follows. We performed a fine discretisation of the field-induced trajectory (Fig. 2b, middle panel), by selecting configurations along this path, which were evenly spaced in their RMSD values. Then, we used each of the considered configurations as a starting point for a MD simulation in absence of the electric field. For each of these simulations we determined the histograms as a function of the RMSD. To estimate the error in the determination of the minima and the barrier between them that arises due to incomplete sampling, we used bootstrapping[114,115]. At each iteration of bootstrapping (from a total of 5000), we removed 20% of each data set and replaced it by a random sampling of the remaining non deleted values.

**Electrostatic potential surface calculations**. The Adaptive Poisson-Boltzmann Solver (APBS) algorithm was used to calculate all potential maps[56] on the PDB 6M0J structural data and on selected frames from the MD trajectories. PDB formats were first prepared by PDB2PQR web server converted to PQR format using CHARMM force field with PROPKA set at pH = 7.0[57]. Thereafter, we carried out the APBS analysis via Linearized Poisson-Boltzmann Equation in VMD software with settings parameters: solvent dielectric constant of 78.5, solvent radius of 1.4 Å, solute dielectric constant of 2.0, system temperature of 300 K, surface density 10.0 points/Å, and using harmonic average smoothing as surface definition.

**Molecular docking calculations**. Protein-receptor interactions were performed using pyDock[58,116] web server, which uses electrostatics and desolvation energy to score docking poses generated with FFT-based algorithms. We approached docking by first selecting the positions and orientations that optimise shape complementarity, followed by a rescoring based on electrostatic, van der Waals and desolvation energies[58]. The first 100 top-scoring structure complexes with the lowest total energy conformations were analysed to evaluate the binding interactions. This docking procedure was applied to the final structure after EF-off in each of the EF intensities evaluated in the shorter sequence. To quantify the likelihood of the docking, we first computed the contacts (as pairs of residues) between the RBD and ACE2 extracted from the 6M0J crystal structure of the RBD–ACE2 complex using a cut-off of 7 Å, measured as arithmetic mean of the atomic distances. We called these contacts native contacts, as they were described elsewhere to be essential for the binding between RBD and ACE2[5,45,52]. From the resulting 100 best docked complexes for each input described above, we extracted the contacts following the same rule as for the native contacts, and selected the six structures that preserve the highest number of the native contacts. We accounted only for the first six structures to yield a comparable distribution of achieved preserved contacts between the different cases, since most of the 100 docked structures in each case showed zero preserved contacts and, therefore, would bias the distribution towards zero if they were included, leaving still recognisable tails but difficult to compare. Since we are interested in the likelihood of a correct binding to exist between a structure under scrutiny and ACE2, discarding the bulk of unmatched cases after the best ones is unlikely to lead to loss of important information about each of the specific structures under comparison. To validate the approach, we repeated the procedure consisting in docking, contacts computation and selection of the best contact preserving structures, for the RBD structures from 6M0J (separated from ACE2 and docked again) and the resulting after thermalisation. As shown above in the main text and Fig. 1, the computational docking of the structures taken from the 6M0J structure almost recover the exact position that was obtained experimentally in two of the cases. The case after thermalisation leads to less contacts than the case of experimental structure, but still around half of the contacts are found more than once, indicating an overall alignment between pairs of residues

that are known to form bonds, and making a clear difference with respect to the analysed cases of structures that were exposed to EF.

**Reporting summary**. Further information on research design is available in the Nature Research Reporting Summary linked to this article.

## Data availability

Raw data of the docking calculations as well as grossly discretised (each ns) MD trajectories can be accessed at the repository https://doi.org/10.5281/zenodo.5153261. The complete raw simulation data are available from the corresponding upon reasonable request. Source data are provided with this paper.

## Code availability

Codes and scripts written for this work as well as instructions for running simulations and analysis tools are provided with this paper (see Supplementary Data 1).

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

## Acknowledgements

Calculations for this research were conducted on the Lichtenberg high-performance computer of the TU Darmstadt, the High Performance Center North (HPC2N) at SNIC, and at local dedicated workstations of our group. M.E.G. and C.R.A. acknowledge support by the PhosMOrg (P/1082 232) research unit of the University of Kassel. P.R. acknowledges the support of the Joachim Herz Foundation through its Add-on Fellowship for Interdisciplinary Life Science. P.R. and M.E.G. received support from Zentrale Forschungsförderung (P2377) of the University of Kassel. C.R.A. thanks the Argentinian AntiCovid Consortium (Argentina) for valuable discussions. C.R.A. and P.R. thank Dr. Bernd Bauerhenne for his help with computing resources. We thank the anonymous reviewers, whose comments led to a substantial improvement of the manuscript.

## Author contributions

M.E.G. devised the main conceptual idea and the project. M.E.G., P.O.-M. and C.R.A. conceived the research plan. P.O.-M. and C.R.A. pre-processed the files for simulations. C.R.A. and P.R. performed the MD simulations. C.R.A. performed detailed identification of changes in structures. C.R.A. and P.R. wrote the software for analysis and analysed the data. All authors analysed and discussed the results and their presentation. All authors wrote and corrected the paper.

## Funding

## Competing interests

A patent application has been filed on 19 March 2021 on an application of the effect described in this paper (Germany patent application WT-JP-4.80.504; patent applicant Universität Kassel). The authors declare no competing interests.
