## [Peer Review File · Nature Communications]

The SARS-CoV-2 Spike Protein is vulnerable to moderate electric fieldsREVIEWER COMMENTS

Reviewer #1 (Remarks to the Author):

The authors have performed MD simulations in the context of explicit solvent at 0.15 M to study the impact of external static electric field (EF) on the secondary and tertiary structures of the segment of S protein SARS-CoV-2 in up conformation (i.e. this segment has residues 319 to 686, which was taken from PDB ID:6VSB after modeling the missing residues, including whole RBD+SD1+SD2 and S1/S2 interface) as well as the native RBD alone from the interface complex of RBD-ACE2 that was obtained from PDB ID:6M0J.

First, they ran MD simulations without EF along 100 ns and 300 ns for the S-protein segment and RBD alone respectively to bring the system to thermodynamic equilibrium and used these atomic coordinates as reference structures. Then, they ran MD simulations along 700 ns with different intensities of EF range from 104 or 105 to 107 V/m as well as higher unrealistic EF (i.e. 109 V/m). In the S-protein segment, they showed a clear evidence that the moderate strengths of EF can induce a structure change as well as reshaped the local interactions. They also checked the stability of the new conformations adopted by S under the external fields but this time they ran extra simulation where the EF is switched off. They showed that the new confirmation states are unaltered after switching off the EF. In the unbound RBD, they showed that the EF also disrupts its secondary structure at several segments, and that its L3 is destabilized from original conformation into open unstructured state. They also discussed the influence of EF on key RBM residues, especially L3, which results in unfavorable positions and orientations of this key residues which implicitly effect the interaction with ACE2. Further, they also compared the electrostatic potential of RBM at different EF cases and they showed that surface charge distribution in the L3 region is greatly affected by EF and this may lead to decrease the electrostatic interactions with ACE2. Finally, they have carried out molecular docking simulation between the EF induced structures of the RBD and ACE2 and showed that the native contacts between them are significantly decreased.

Overall, this study systematically investigates the effect of EF on the S-protein and RBD, providing new insights on how EF can be used to reduce SARS-CoV-2 infectivity by inducing structural conformation and rearrangement of local interactions in the RBM residues. The approach used can help other calculations related to mutation in S-protein and in other biological system. The construction of models is appropriate, the force fields chosen is adequate and the graphics are truly outstanding. Overall, the paper is well-written and well-referenced. This manuscript is publishable after following suggested revisions.

Major point:

The MD study is limited to unbound RBD where the key residues of RBM are freely to fluctuate during simulations in both EF cases (with or without). I suggest running new MD simulations for bound RBD-ACE2 complex and use only one EF strength (i.e. 106 V/m) with less MD timescale (let say 100 ns and 400 ns for without and with EF). If possible, applied the EF only on the RBD in the bound state. Then, calculate the binding free energy in both cases and check the dissociation between RBD and ACE2 in the case of EF-on. This will make the paper to have more explicit solid conclusion that the EF has a significant effect on the binding between RBD and ACE2.

Minor points:

1. In the Method section, the authors should explain how they applied the EF in some more details. What is the direction of this EF? If it is applied only in one direction only, some sentences in the main text should be added to point out the drawback of this choice because RBD is asymmetric and EF changes in the direction will affect the charge distribution and the permanent dipole.
2. I recommend excluding any citations in the abstract.

Reviewer #2 (Remarks to the Author):

This is a potentially interesting paper on the effect of an electric field on the spike proteins of SARS-CoV-2. The idea is that the field is responsible for non-specific inactivation by disabling the bond to the receptor.

I have a few concerns with the current version of the manuscript which I feel need to be addressed.

1. It is not too convincing to accept that the field-induced effects found by simulations are realistic without a control simulation, which shows no or little effect for fields of similar magnitude for analogous proteins in other coronaviruses, less sensitive to the field, or at least for other simpler proteins. If the electric field is large enough to lead to conformational changes when simulating most proteins then the case for novelty and interest would be significantly diminished. It could be that these calculations exist already but the authors would need to discuss them in that case.
2. An important aspect is that the field-induced configurations are metastable also after field removal. It would be good to show a free energy profile which combines both the off-field and the field-induced configurations. If I am not mistaken the profiles shown in Fig. 2b never compare directly off-field and field-induced configuration (i.e., they do not span both states).
3. Polarisable force fields are important for calculations like these, yet polarisation effects are notoriously tricky to accurately capture. I think the authors should discuss more at least in the method section the validation for calculations at the field used here.
4. In the discussion of Fig. 3, the authors identify some key residues which are important for the interactions with the receptor and which are particularly affected by the field. This discussion would be more convincing if coupled to simulations of mutants without one or more of these key residues, where the effects of mutations on interaction with receptor, field-induced configurations and Poisson-Boltzmann electrostatics are shown. I think these simulations are in principle doable and I think they would strengthen the manuscript.

Reviewer #3 (Remarks to the Author):

In this manuscript Arbeitman et al. report an interesting study of the effect of electrical fields on the stability of the SARS-CoV-2 protein. This study would be the origin of an important approach to block the infection process of this virus. The manuscript is well written, and the results are clearly exposed. However, I have some concerns about some issues that do not seem clear enough.

There is a lack of details about how the EF is applied. In particular, the proteins are submerged in aqueous solution, where the electrical fields are screened by the dielectric constant of water. In principle, this would decrease the strength of the electrical field in about 100. Therefore, the strength of the electrical field to be applied externally should be about 100 times larger. Is this value still in the "low" electrical field range?

The simulations reported in this manuscript clearly show that the protein structure is irreversibly destroyed by the electrical field. For the sake of completeness, it is necessary to show negative examples, i.e. proteins which are not affected by these electrical fields. For example, the authors can take other virus spikes (flue, human adenovirus, etc) to show whether their fibers are also affected. I do not have any doubt that this methodology would be a great discovery to stop the infection of many viruses with spikes, but this theory has to be falsify with other systems.

Reply to the Reviewers

We thank all three reviewers for their positive comments regarding our work and for their constructive criticism, which, in our opinion, has led to a much stronger article.

We have revised our manuscript taking into account all suggestions and requirements of the reviewers. Below, we address all points raised by the reviewers and indicate the corresponding changes that we have made on the manuscript.

Reviewers' comments are copied in blue for clarity. Our answers are written in black, and the new texts of the manuscript are indicated in green and under quotation marks. Additional References are named here as RRn (n=1,24) in order of appearance. In the manuscript, we have incorporated them and given them the proper number. Due to the additional references, we have rearranged the numeration throughout the manuscript.

To simplify the tracking of changes in the revised manuscript we attach, together with the original revised version, a copy in which changes with respect to the previous version are highlighted (yellow background).

Reply to REVIEWER 1

Major point:

The MD study is limited to unbound RBD where the key residues of RBM are freely to fluctuate during simulations in both EF cases (with or without). I suggest running new MD simulations for bound RBD-ACE2 complex and use only one EF strength (i.e. 106 V/m) with less MD timescale (let say 100 ns and 400 ns for without and with EF). If possible, applied the EF only on the RBD in the bound state. Then, calculate the binding free energy in both cases and check the dissociation between RBD and ACE2 in the case of EF-on. This will make the paper to have a more explicit solid conclusion that the EF has a significant effect on the binding between RBD and ACE2.

We appreciate the reviewer's comment and observation. Following the above suggestion, we have performed additional simulations of the bound RBD-ACE2 complex in which only the RBD is subject to an electric field of magnitude 10^6 V/m. However, it is important to point out that this condition differs from the starting hypothesis and main focus of our manuscript, in which the application of an **EF directly to the S protein or possibly to the isolated virion** is proposed with the aim of producing severe conformational changes that **then** affect recognition or binding to the receptor. The required simulations took some time due to the larger size of the system compared to the previous ones considered. We used the PDB structure ID:6M0J as the starting structure for molecular dynamics simulations on the RBD-ACE2 complex.

In order to avoid that the EF affects the ACE2 receptor we applied position restraints on the backbone atoms (with a strong force constant of $100000.0 \text{ kJ mol}^{-1} \text{ nm}^{-2}$) and also on the heavy atoms of the side chains of ACE2 (with a force constant of $10000.0 \text{ kJ mol}^{-1} \text{ nm}^{-2}$). In this way, the EF only affects the RBD in the bound state. Initial simulations with no-EF were performed during 300 ns, and

then an EF of strength 10^6 V/m was applied during 500 ns. Results of those constrained simulations confirm our previous conclusions on the isolated RBD, namely, that the main EF induced damage occurs in the loop 3 (L3) region (see Extended Data Fig. 8 below). The EF of strength 10^6 V/m induces a transition of the well-defined secondary structures (β -strands) of L3 into an unstructured loop. This means that, even if the structure of the RBD is confined due to the bound state with ACE2, the EF induces small relative motions which add up to a dramatic conformational change. Moreover, partial dissociation from the receptor was detected within the 500 ns of the simulation. As shown in Extended Data Fig. 8, the change of L3 to an unstructured coil suggests that residues adjacent at the loop region suffer a local reorganisation and thereby alter the molecular architecture of the RBD–ACE2 interface. We have addressed this issue in the Discussion and in the Extended Data Figure 8, where we explain details in the caption (see below).

Text in Discussion

“In Extended Data Fig. 8 we also show that the EF produces structural damage in the RBD even when it is bound to the ACE2 receptor.”

Extended Data Figure 8: EF induces damage on the RBD even under confinement inside the RBD-ACE2 complex. We considered the bound RBD-ACE2 complex and applied an external electric field only to the RBD. This was achieved by implementing position restraints on the back

bone atoms and on the heavy atoms of the side chains of ACE2, using large force constant values of $100000.0 \text{ kJ mol}^{-1} \text{ nm}^{-2}$ and $10000.0 \text{ kJ mol}^{-1} \text{ nm}^{-2}$, respectively. In this way, the EF only affects the RBD in the bound state. Initial simulations with no-EF were performed during 300 ns, and then an EF of strength 10^6 V/m was applied during 500 ns. Results of those constrained simulations confirm our previous conclusions on the isolated RBD, namely, that the main EF induced damage occurs in the loop 3 (L3) region before and after application of an EF of 10^6 V/m (only on the RBD). The figure shows the full view of the secondary structures of the complex before and after EF application. The orientation of the ACE2 receptor is the same in both figures. The L3 loop in each of the structures is highlighted with a circle. The disappearance of the two β -sheets in the L3 loop of the RBM and a partial RBD-ACE2 dissociation can be clearly observed on the right panel. In comparison with no-EF, the RBM is not as tightly associated after application of an EF.

Minor points:

1. In the Method section, the authors should explain how they applied the EF in some more details. What is the direction of this EF? If it is applied only in one direction only, some sentences in the main text should be added to point out the drawback of this choice because RBD is asymmetric and EF changes in the direction will affect the charge distribution and the permanent dipole.

We thank the reviewer for pointing out this issue (also addressed by REVIEWER 3). We agree with both reviewers that a detailed description of the EF application was indeed missing in the previous version. We apologize for that. We have added a corresponding text in the Methods' section. We apply the electric field in a predefined direction (x-direction in our case) with respect to the simulation cell. However, this does not represent a restriction, since the considered protein segments are allowed to rotate freely. The EF-on simulations are long enough to allow the protein to rotate in different directions, so that on a time-average there is no preferred axis. We confirmed this fact by visual inspection of the output files. Therefore, the protein is, in fact, successively affected by fields in all directions. Notice that this resembles a real situation in which the virus rotates in the space between two electrodes where the field is acting.

We have added a subsection "Application of the electric field" to the Methods' section. The complete text of the subsection is given below and includes also the points raised by REVIEWER 3:

"Application of the electric field"

The electric fields were applied in x-direction with respect to the MD simulation box. The electric field couples to all charges in the system, including the charged atoms in the protein, in the water molecules and the isolated Na^+ and Cl^- ions. The interaction of the system with the external static electric field $E = (E_0, 0, 0)$ is introduced by an additional force term of the form $F_i = Eq_i$ acting on

atom i in the MD cell, where q_i refers to the atomic charge. The protein is allowed to freely rotate. During the production runs the protein rotated around many different axes, as was confirmed by visual inspection of the results. This means that there is no preferred direction for the application of the electric field on the protein. Note that this resembles a real situation in which the virus rotates in the space between two electrodes where the field is acting. “

A further piece of text was added to this paragraph in the manuscript in response to the points of REVIEWER 3 (see the corresponding reply below).

2. I recommend excluding any citations in the abstract.

We followed the recommendation of the reviewer and shifted the citations in the abstract to the introduction.

Reply to REVIEWER 2

1. It is not too convincing to accept that the field-induced effects found by simulations are realistic without a control simulation, which shows no or little effect for fields of similar magnitude for analogous proteins in other coronaviruses, less sensitive to the field, or at least for other simpler proteins. If the electric field is large enough to lead to conformational changes when simulating most proteins then the case for novelty and interest would be significantly diminished. It could be that these calculations exist already but the authors would need to discuss them in that case.

We thank the reviewer for pointing out this important aspect, also addressed by REVIEWER 3. It is certainly important to stress the fact that the spike protein is, by far, more sensitive than most other proteins to the influence of external electric fields by providing examples. Instead of performing an additional simulation by ourselves on a particular simple protein chosen ad-hoc, we prefer to provide in the paper a list of examples from simulations performed and published previously by other groups on the same level of accuracy, i.e., using the same or equivalent force fields, using the same description of the water molecules, a similar simulation length, and coupling the electric field microscopically at the same theoretical level or even at a Density-Functional-Theory level). Some selected examples are given below (two of them were actually already cited in the previous version, though not discussed in this context). It is important to point out, that, so far, studies using molecular dynamics simulations to address the structural response of proteins to electric fields have reported **no changes** in the **secondary structure** of proteins and peptides for electric field intensities **below** 10^8 V/m. For instance, Wang et al., [RR1] reported that the secondary structure of **insulin** remains intact under external electric field intensities equal or below 0.15 V/nm ($1.5 \cdot 10^8$ V/m), and it becomes

disrupted for values equal or greater than 0.25 V/nm. Similar results were obtained by Budi et al., [RR2]. Astrakas et al. [RR3] studied the response of **chignolin** to an electric field and found that strengths of 0.1 V/nm (10^8 V/m) did not affect chignolin's conformation, a strength of 0.25 V/nm (2.5×10^8 V/m) causes mild effects, while electric fields of intensities 0.5 V/nm and 1 V/nm induce significant conformational changes. Marrachino et al., [RR4] used **myoglobin** as their case study, and found no distinction between application of a field of strength 10^8 V/m and no field condition. In contrast, a field intensity of 10^9 V/m produced the unfolding of the protein. Ilieva et al., [RR5] used density functional theory calculations in combination with a polarisable continuum model to study the influence of electric fields on **helical structures** of peptides, and found an electric field strength of 2.5 V/nm as the threshold value for disruption of the helical structure. Moreover, della Valle et al., [RR6] reported a threshold of 5×10^8 V/m for relevant conformational changes when applying pulsed electric fields to **superoxide dismutase**. Values of the same order of magnitude were reported in Refs. [RR7] on **dihydrolipoamide succinyltransferase**, [RR8] and [RR9] on the **amyloidogenic apolipoprotein apoC-II**.

As shown above, there is an overwhelming evidence that the numerically predicted damage thresholds for most proteins under application of electric fields for short times (of the order of microseconds) is higher than 10^8 V/m. This justifies our conclusion that the spike protein of SARS-CoV-2 and especially its RBD are unusually vulnerable to electric fields.

[RR1] Wang, X., Li, Y., He, X., Chen, S. & Zhang, J. Z. H. Effect of Strong Electric Field on the Conformational Integrity of Insulin. *The Journal of Physical Chemistry A* **118**, 8942–8952 (2014)].

[RR2] Budi, A., Legge, F. S., Treutlein, H. & Yarovsky, I. Electric Field Effects on Insulin Chain-B Conformation. *The Journal of Physical Chemistry B* **109**, 22641–22648 (2005).

[RR3] Astrakas, L., Gousias, C. & Tzaphlidou, M. Electric field effects on chignolin conformation. *Journal of Applied Physics* **109**, 094702 (2011).

[RR4] Marracino, P., Apollonio, F., Liberti, M., d'Inzeo, G. & Amadei, A. Effect of High Exogenous Electric Pulses on Protein Conformation: Myoglobin as a Case Study. *The Journal of Physical Chemistry B* **117**, 2273–2279 (2013).

[RR5] Ilieva, S., Cheshmedzhieva, D. & Dudev, T. Electric field influence on the helical structure of peptides: insights from DFT/PCM computations. *Phys Chem Chem Phys* **21**, 16198–16206 (2019).

[RR6] della Valle, E., Marracino, P., Pakhomova, O., Liberti, M. & Apollonio, F. Nanosecond pulsed electric signals can affect electrostatic environment of proteins below the threshold of conformational effects: The case study of SOD1 with a molecular simulation study. *PLOS ONE* **14**, e0221685 (2019).

[RR7] Jiang, Z.; You, L.; Dou, W.; Sun, T.; Xu, P. Effects of an Electric Field on the Conformational Transition of the Protein: A Molecular Dynamics Simulation Study. *Polymers* **11**, 282 (2019).

[RR8] Todorova, N., Bentvelzen, A., English, N. J. & Yarovsky, I. Electromagnetic-field effects on structure and dynamics of amyloidogenic peptides. *The Journal of Chemical Physics* **144**, 085101 (2016).

[RR9] Todorova, N., Bentvelzen, A. & Yarovsky, I. Electromagnetic field modulates aggregation propensity of amyloid peptides. *The Journal of Chemical Physics* **152**, 035104 (2020).

We have condensed the above discussion both in the introduction and in section Discussion and added some of the above mentioned references.

Sentence in the Introduction:

“So far, studies using molecular dynamics (MD) simulations to address the structural response of proteins to electric fields applied for around 1 microsecond have reported no changes in the secondary structure of proteins and peptides for electric field intensities below a field strength of 10^8Vm^{-1} [25,29-32]. In this work, we show, via MD simulations, that EFs of much lower intensities ($10^5 - 10^7 \text{Vm}^{-1}$) cause, on a sub-microsecond time scale, significant damage on the tertiary and secondary structure of the S protein that affects its interaction with ACE2, potentially making SARS-Cov-2 less infectious.”

Sentences in the Discussion

“Summarising, this study demonstrates that EFs of different biologically relevant strengths change S of SARS-CoV-2 both at nanometre and sub-nanometre scales. Considerable changes in the secondary structure of the RBD in the wild type and currently dominant mutants of S occur at field strength orders of magnitude smaller than for most proteins. We conclude that the spike protein of SARS-CoV-2 and especially its RBD are unusually vulnerable to external electric fields. “

2. An important aspect is that the field-induced configurations are metastable also after field removal. It would be good to show a free energy profile which combines both the off-field and the field-induced configurations. If I am not mistaken the profiles shown in Fig. 2b never compare directly off-field and field-induced configuration (i.e., they do not span both states).

The reviewer is right. In the middle panel of Fig. 2b we show both the initial and the “damaged” states, but both under the influence of the electric field. In order to allow for a direct comparison between the states shown in the upper panel (before field) and in the lower panel (after field) of Fig. 2b, we proceeded as follows. We performed a fine discretisation of the field induced trajectory, picking up configurations along this path. Then, we used each of the considered configurations as a starting point for a MD simulation in absence of the electric field. For each of these simulations we determined the histograms as a function of the RMSD as it was explained in the section Methods in the manuscript. Moreover, we used bootstrapping [RR10] and [RR11] to estimate the error at the minima and in the barrier between them, that arises due to incomplete sampling. At each iteration of bootstrapping (from a total of 5000), we removed 20% of each data set and replaced it by a

random sampling of the remaining non deleted values. The results are summarized in the new Extended Data Fig. 3 and also discussed in the main text. From Extended Data Fig. 3, the existence of the barrier is clear and the minimum after field application is not simply a rapidly decaying metastable state but rather a quite stable, long-lasting state. This supports our previous conclusions. We thank the reviewer for the hint.

[RR10] Efron, B. Bootstrap methods: Another look at the jackknife. *The Annals of Statistics*. **7** (1), 1–26 (1979).

[RR11] Demuynck, R. et al. Efficient construction of free energy profiles of breathing metal-organic frameworks using advanced molecular dynamics simulations. *J. Chem. Theory Comput.* **13**, 5861–5873 (2017).

Extended Data Figure 3: estimate of the free energy profile (EF 10^6 V/m) combining the off-field (EF-off) and the field induced configurations *in absence* of electric fields. The error bars were obtained by means of bootstrapping with replacement (see Methods).

We added the above figure to the Extended Data and discussed it in the main text and in section Methods.

In the main text:

“Notice that the middle panel of Fig. 2b shows both the initial and the “damaged” states, but both under the influence of the electric field. We have also determined the free energy profile connecting both states in absence of fields (see Methods Extended Data Fig. 3). A barrier separating both states can be clearly observed. Moreover, the figure shows that the minimum after field application is not simply a rapidly decaying metastable state but rather a quite stable, long-lasting state.”

In Methods:

“The free energy profile in Extended Data Fig. 3 was generated as follows. We performed a fine discretisation of the field induced trajectory (Fig. 2b, middle panel), by selecting configurations along this path, which were evenly spaced in their RMSD values. Then, we used each of the considered configurations as a starting point for a MD simulation in absence of the electric field. For each of these simulations we determined the histograms as a function of the RMSD. To estimate the error in the determination of the minima and the barrier between them that arises due to incomplete sampling, we used bootstrapping [RR10,RR11]. At each iteration of bootstrapping (from a total of 5000), we removed 20% of each data set and replaced it by a random sampling of the remaining non deleted values.”

3. Polarisable force fields are important for calculations like these, yet polarisation effects are notoriously tricky to accurately capture. I think the authors should discuss more at least in the method section the validation for calculations at the field used here.

Following the suggestion of the reviewer we performed, for the sake of validation of our calculations, short-time simulations (0.2ns) of the RBD in the presence of an electric field by using the Drude polarisable force field [R12,R13,R14]. These simulations were performed on a non-glycosylated RBD, since glycosylated residues are not supported by the polarisable force field in our current workflow which uses NAMD, CHARMM, CHARMM-GUI. However, the most important features of the protein-EF interaction should not be perturbed by this approach because glycosylated residues are far enough from the region of interest (RBM). We considered different values of the EF: 0 V/m, 10^4 V/m, 10^5 V/m, and 10^6 V/m and found, in agreement with the previous calculations in the manuscript, using the CHARMM-36 standard force field, that the RBM is affected by the EF even during the short simulation time of 0.2ns considered here. For instance, the EF induced changes are reflected in a temporal suppression of the spatial fluctuations in some residues and the amplification of the fluctuations on others (e.g., around residues 476 and 499). In Extended Data Fig. 10 we show this behaviour. For comparison, we also plot in Extended Data Fig. 10 the root-mean square fluctuations of the residues in the RBM using the standard force field for a simulation time of 1ns. The effect of temporal suppression and amplification of fluctuations can be also observed, though not always in the same amino acids on this short time-scale. However, and most importantly, the magnitude of the fluctuations is similar.

Notice that long-time simulation using the polarisable force field would be computationally not affordable, because the time-step needs to be extremely small (0.1 fs) to avoid instabilities in the simulations. Nevertheless, and in spite of the ultrashort time-step used, we detected instabilities in some of the trials and upon restarting the simulations. In conclusion, results of Extended Data Fig.

10 confirm the general trend originally obtained in our work and, in our opinion, validate the calculation with the standard (CHARMM-36) force field.

[R12] Lamoureux, G., Harder, E., Vorobyov, I. V., Roux, B., and MacKerell, A. D. 2006. A polarizable model of water for molecular dynamics simulations of biomolecules. *Chem. Phys. Lett.* , **418**, 245 (2006).

[R13] Yu, H., Whitfield, T. W., Harder, E., Lamoureux, G., Vorobyov, I., Anisimov, V. M., MacKerell Jr., A. D., and Roux, B. 2010. Simulating Monovalent and Divalent Ions in Aqueous Solution Using a Drude Polarizable Force Field. *J. Chem. Theory Comput.* **6**, 3, 774 (2010).

[R14] Lopes, P. E. M., Huang, J., Shim, J., Luo, Y., Li, H., Roux, B., and MacKerell, Jr., A. D. 2013. Polarizable Force Field for Peptides and Proteins Based on the Classical Drude Oscillator. *J. Chem. Theory Comput.*, **9**, 12, 5430 (2013).

We have included the following text in the section Methods:

“Validation of the force field

For the sake of validation of the non-polarisable CHARMM-36 force field, simulations with the Drude polarisable force field [R12,R13,R14] were performed using the CHARMM-GUI and CHARMM (v 43a1) software. The initial structure was the same as for the simulations using the CHARMM-36 force field except for the glycosylated residues which were modelled without sugar rings. The cubic box had an initial volume of $(100\text{\AA})^3$ and a final size of $(97.8\text{\AA})^3$ after equilibration. These simulation sets were run with the NAMD software (v. 2.14). The resulting system consisted of 156,541 atoms including 86 Na^+ and 93 Cl^- ions. Electrostatic interactions were solved with the Particle Mesh Ewald method with a grid spacing of 1.5 and splines 6th interpolation order. A Drude temperature of 1 K was used, with a damping -coefficient of 20 ps^{-1} , a bond length of 0.2 \AA , and a Drude force constant of $40,000\text{ kcal/mol \AA}^2$. Also, a non-bonded Thole interaction radius of 5 \AA together with a Drude hard wall option were considered. Initially, 10000 minimisation steps were done to avoid atomic clashes followed by a 1 ns equilibration run by using the Langevin thermostat with a damping coefficient of 5ps^{-1} including the hydrogen atoms. Pressure was controlled using the Nose-Hoover Langevin barostat with a target pressure of 1.01 bar, piston period of 50 fs and piston decay of 25 fs. The time step and temperature were 0.1 fs and 303.15 K, respectively. The production runs under EF application were 0.2ns long.

Extended Data Fig. 10 shows the comparison of the results for the root mean square fluctuations inside the RBM for different EF strengths using both force fields. The similar results, particularly concerning the magnitude of the fluctuations, confirm that the force field CHARMM-36 is accurate enough for this kind of simulation. Notice that long-time simulation using the polarisable force field

would be computationally not affordable within a reasonable computer time due to the ultrashort time-step needed (100 attoseconds). Moreover, polarisable force fields exhibit instabilities”

Extended Data Figure 10: root mean square fluctuations of the RBM (wild type) for different EF strengths using a polarisable and a non-polarisable force field. Upper panel: Drude polarisable force field (simulation over 0.2ns). Lower panel: CHARMM-36 force field (simulation over 1ns).

4. In the discussion of Fig. 3, the authors identify some key residues which are important for the interactions with the receptor and which are particularly affected by the field. This discussion would be more convincing if coupled to simulations of mutants without one or more of these key residues, where the effects of mutations on interaction with receptor, field-induced configurations and Poisson-Boltzmann electrostatics are shown. I think these simulations are in principle doable and I think they would strengthen the manuscript.

We would like to thank the reviewer for this important suggestion. We took this useful comment seriously, but instead of considering artificial mutants, we decided to analyse the effect of electric fields on real mutants of SARS-CoV-2, namely, the three currently circulating variants of concern B.1.1.7 (UK), B.1.351 (South Africa) and P.1 (Manaos, Brazil). These additional calculations clearly show that external electric fields of strengths as low as 10^5 V/m produce severe damage in the RBD of the three variants. We have added a new section containing a new figure (Fig. 4) to the manuscript addressing these calculations. Furthermore, we added a figure to the Extended Data section (Extended Data, Fig. 9). Finally, we briefly updated the discussion correspondingly. The full text of the additional section as well as both figures are presented below:

New Section:

“RBD mutations corresponding to the variants of concern B.1.1.7 (UK), B.1.351 (South Africa) and P.1 (Brazil) are effectively damaged by moderate electric fields

The recently emerged mutations of the S protein in the new “variants of concern” (VOC) 501Y.V3/P.1 (Brazil), 501.V2/ B.1.351 (South Africa) and 501Y.V1/B.1.1.7 (UK) generate great worries because they exhibit a significantly higher infection rate and circulate globally [R15,R16,R17,R18]. Important consequences of the circulation of the emerging variants are increased transmissibility, pathogenicity and ability to escape from neutralising antibodies or vaccine-induced response [R19,R20]. The multiple mutations in the VOC also affect the RBD through critical residues as N501Y, E484K, K417N and K417T [R21] which are believed to enhance the interactions between the spike protein of SARS-CoV-2 and the ACE2 receptor [R22,R23]. In this work, we have also studied the effect of EFs on the RBD of the VOC. As in the case of the wild-type, we considered as a starting structure of the RBD the chain E from the PDB structure ID: 6M0J (see Methods), which we used as a template. We then generated the three VOC by the corresponding amino acid exchanges. We performed different runs (thermalisation no-EF 500 ns, EF-on 10^5 V m⁻¹ 1000 ns and EF-off 300 ns) following the previously applied protocol (see Methods). Remarkably, the secondary structure of the

L3 loop of the three considered VOC also undergoes a transition from the closed structure with the two beta-sheets to an open unstructured coil. Subsequent EF-off simulations for 300 ns revealed no differences in the unstructured nature of L3. In Fig. 4 we show the corresponding RBD-structures and RBM-electrostatic potentials in absence, under the presence, and after switch-off of an external field with a strength as low as 10^5 Vm^{-1} . This shows that the external electric field has a significant impact on RBD of the mutants which leads to conformational changes in the region localized near the ACE2 interaction interface. These changes are, at least, as serious as for the wild-type (s. Fig. 4). The structural implications of the VOC mutations on the electrostatic potential at the interface between the RBD of the spike protein and ACE2 for the final structures (after no-EF, EF-on, and EF-off runs) were evaluated by solving the Poisson-Boltzmann equations (APBS package). As shown in Fig. 4, the spatial distribution of the electrostatic potential all over the RBM is strongly affected in all three variants upon EF application. The effect of the E484K mutation in the B.1.351 and P.1 variants manifest itself in a large-amplitude motion and expulsion of Lys484 from the central hydrophilic patch at the interface to ACE2, altering the electrostatic binding. Note, that this movement causes strong rearrangement of the residues and, consequently, changes in side chain conformation, which should also alter the contact numbers with ACE2. Summarising, our simulations on the VOC predict that for all three considered mutants, an external EF as low as 10^5 V m^{-1} produces a severe structural damage and conformational reorganisation of the RBD–ACE2 interface.”

[R15] <https://www.ecdc.europa.eu/en/covid-19/variants-concern>, 19 May 2021.

[R16] Wise, J. Covid-19: new coronavirus variant is identified in UK. *BMJ* **371**, m4857 (2020).

[R17] Naveca, F. G. *et al.* COVID-19 in Amazonas, Brazil, was driven by the persistence of endemic lineages and P.1 emergence. *Nat Med* <https://doi.org/10.1038/s41591-021-01378-7> (2021).

[R18] Tegally, H. *et al.* Detection of a SARS-CoV-2 variant of concern in South Africa. *Nature* <https://doi.org/10.1038/s41586-021-03402-9> (2021).]

[R19] Wang, P. *et al.* Antibody resistance of SARS-CoV-2 variants B.1.351 and B.1.1.7. *Nature* <https://doi.org/10.1038/s41586-021-03398-2> (2021).

[R20] Dejnirattisai W. *et al.* Antibody evasion by the P.1 strain of SARS-CoV-2. *Cell* <https://doi.org/10.1016/j.cell.2021.03.055> (2021).

[R21] Socher, E. *et al.* Mutations in the B. 1.1. 7 SARS-CoV-2 spike protein reduce receptor-binding affinity and induce a flexible link to the fusion peptide. *Biomedicines* **9**, 525 (2021).

[R22] Singh, J. *et al.* Structure-Function Analyses of New SARS-CoV-2 Variants B. 1.1. 7, B. 1.351 and B. 1.1. 28.1: Clinical, Diagnostic, Therapeutic and Public Health Implications. *Viruses* **13**, 439 (2021).

[R23] Starr, T. N. *et al.* Deep mutational scanning of SARS-CoV-2 receptor binding domain reveals constraints on folding and ACE2 binding. *Cell* **182**, 1295–1310 (2020).

Figure 4. Mutants of the RBD corresponding to the variants of concern B.1.1.7 (UK), B.1.351 (South Africa) and P.1 (Brazil) are also irreversibly damaged by electric fields. The three variants were generated by replacing individual residues *in-silico* in the 6M0J structure. The introduced mutations were N501Y (P.1, B.1.1.7 and B.1.351) K417N (B.1.351), K417T (P.1) and E484K (B.1.351 and P.1). For each of the variants, the secondary structures and the electrostatic potential surfaces are shown for the conditions no-EF, EF-on (10^5 Vm^{-1}) and EF-off, in the left, middle and right columns,

respectively (details in Methods and caption of Figure 3). The position of mutated residues is shown on the electrostatic potential surfaces. **Upper panel**, results for variant B.1.1.7 (UK). **Middle panel**, results for variant B.1.351 (South Africa). **Lower panel**, results for variant P.1 (Brazil). In all variants, the secondary structure is severely disrupted by the electric field at the level of the two parallel β sheets in the loop L3, that turns to an unstructured coil. The electrostatic potential surfaces change under the EF to a distorted pattern in analogous way as in the wild type (see Figure 3).

Extended Data Figure 9: Structural flexibility of the RBD of the variants of concern B.1.1.7, B.1.351 and P.1 under EF. Root mean square fluctuations (RMSF) of the RBM region in the VOC in absence and presence of an external field of strength 10^5Vm^{-1} was applied. Even in absence of fields the RBM region shows an increased flexibility in the three variants when compared to wild type. Residues of the RBD that considerably increase their flexibility under EF application, particularly S477, E484 and K484, are critical and situated at the interface region to the ACE2 receptor.

Reply to REVIEWER 3

There is a lack of details about how the EF is applied. In particular, the proteins are submerged in aqueous solution, where the electrical fields are screened by the dielectric constant of water. In principle, this would decrease the strength of the electrical field in about 100. Therefore, the strength

of the electrical field to be applied externally should be about 100 times larger. Is this value still in the “low” electrical field range?

As we pointed out in the reply to REVIEWER 1, we apologize for not describing the coupling to the electric field in detail in the previous version. In the revised version we dedicate a subsection of Methods to describe the interaction of the system with the field. We thank the referee for the question regarding the dielectric response of the solution, because this is an important point which must be clearly presented in the manuscript. The screening of water **is microscopically included** in the simulations, since the field couples to all charged particles in the system, including ions in the solution and the H- and O- atoms of the water molecules (see reply to REVIEWER 1). Consequently, the water molecules become polarized, generating an electric field opposite to the applied one (screening). Therefore, the protein is indeed affected by an effective field composed by the vectorial sum of the external electric field and the induced field due to the water polarisation. The effective field experienced by the protein is therefore considerably smaller than the applied field corresponding to the values we give in the manuscript. The microscopic description of the effect of water polarisation in MD simulations was nicely described by Caleman and van der Spoel in Ref. [R24] (Fig.1 of the Supporting Information). The description in terms of a dielectric constant is a coarse-grained view of this effect with further assumptions, like linear response, but it leads to similar results. However, the microscopic treatment is more accurate. The bottom line of the above discussion is: the field strengths mentioned in the paper represent the **applied external field** and not the effective field acting on the protein. In the new subsection “**Application of the electric field**” in Methods we added the following text passage (after the text presented in the reply to REVIEWER 1):

“The EF strengths mentioned throughout this work refer to the applied external field. Since we consider the coupling of all charged atoms, including ions in the solution and the H- and O- atoms of water, with the EF, the water molecules become polarized, generating an electric field opposite to the applied one, effectively screening it. Therefore, the protein is indeed affected by a total field composed by the vectorial sum of the external electric field and the induced field due to the water polarisation[R24].”

[R24] Caleman, C and van der Spoel, D.. Picosecond Melting of Ice by an Infrared Laser Pulse: A Simulation Study. *Angew. Chem.* **120**, 1439 –1442 (2008)

The simulations reported in this manuscript clearly show that the protein structure is irreversibly destroyed by the electrical field. For the sake of completeness, it is necessary to show negative examples, i.e. proteins which are not affected by these electrical fields. For example, the authors can take other virus spikes (flue, human adenovirus, etc) to show whether their fibers are also affected. I do not have any doubt that this methodology would be a great discovery to stop the

infection of many viruses with spikes, but this theory has to be falsified with other systems.

First of all, we thank the referee for the positive assessment of our proposed methodology. As mentioned in our reply to REVIEWER 1, we now provide in the revised version enough evidence, based on previous careful studies on the same level of accuracy, that most proteins can only be severely damaged by electric fields of high intensities. We have added the corresponding text passages in the introduction and discussion. Our point, that the Spike protein is more vulnerable to electric fields than most other proteins is, in our opinion, confirmed. The fact that even fields of intensity as low as 10^5V/m and duration of 1 microsecond destroy the secondary structure of the L3 loop, as shown by the additional simulations on the variants of concern, also point in the same direction. We expect a similar behaviour for other viruses containing spike proteins, like human adenovirus. This will be the subject of future studies in our group and we hope that this manuscript can motivate experimental groups to start joint collaborations.

As mentioned above, we have added sentences in the introduction and discussion regarding this issue.

REVIEWER COMMENTS

Reviewer #1 (Remarks to the Author):

I read the revised version for this submission and am satisfied with authors response to all three reviewers. The reply to all three reviewer is very detailed, and in some cases beyond the expectation. For example, the authors added new resulted related to the effects on the new variants in SARS-CoV-2 Spike protein. I recommend the paper be accepted in the present form.

Reviewer #3 (Remarks to the Author):

I am in general satisfied with the authors responses. However, I still have doubts about the way they compare the damage of the coronavirus spikes by EFs with other proteins. They included references of other groups working in proteins, but did not discuss the action of low EFs in other viruses' spikes or appendages. They do not say what is special about the structure or architecture that makes the spikes prone to be damaged by low EFs, compared with other virus appendages. Indeed, they nicely include new virus variants, and seems that whatever makes these spikes sensible to low EFs, is conserved. My question is: what is conserved?

Reply to REVIEWER 3

We really appreciate the constructive criticism of reviewer 3, which triggered an interesting opinion exchange among the authors finally leading to an improved version of the discussion of results.

Reviewer 3:

I am in general satisfied with the authors responses. However, I still have doubts about the way they compare the damage of the coronavirus spikes by EFs with other proteins. They included references of other groups working in proteins, but did not discuss the action of low EFs in other viruses' spikes or appendages. They do not say what is special about the structure or architecture that makes the spikes prone to be damaged by low EFs, compared with other virus appendages. Indeed, they nicely include new virus variants, and seems that whatever makes these spikes sensible to low EFs, is conserved. My question is: what is conserved?

The reviewer poses a key question that helps to better understand our results and put them in a proper context. To the best of our knowledge, no previous studies (neither theoretical nor experimental) have addressed the effects of EF on the spike protein or any other viral membrane protein. Therefore, it is not possible at the moment to make a reasonable comparison. Taking into account this fact, our comparative analysis was based on structural characteristics that have been observed in either functionally similar proteins without EF or structurally different proteins under EF. In view of the lack of studies of the effect of EF on viral proteins, we believe that our work helps introducing a field of structural biology that is worthy of exploring in future investigations.

Regarding the second remark and the final question of the reviewer, we believe that we have found a plausible answer. The spike protein, similarly to other class I viral fusion proteins such as influenza virus hemagglutinin, gp120-gp41 HIV, or Ebola virus GP, mediates fusion by a process involving dramatic conformational changes to lower energy states, where the post-fusion conformation is more stable at physiological conditions [Refs.1-7]. This means that the prefusion state of those proteins is metastable [Refs.8-10], which seems to be important for optimizing or regulating their functions [Refs.11-19]. Thus, function of viral fusion proteins depends on their ability to fold into a less stable but functionally relevant pre-fusion conformation, while preventing the transition to the more stable post-fusion conformation. Therefore, the kinetic accessibility of the post-fusion state is limited before docking to the ACE2 receptor [Refs.20-21]. Now, the energy barrier trapping the prefusion state was found to be surprisingly low in the case of coronaviruses spike proteins [Refs.22-26]. *We hypothesise that low to moderate electric fields are capable enough to supply the protein enough energy to produce a non-thermal energy barrier crossing to a state between the prefusion and the post-fusion state. We also expect this effect to occur in other viral fusion proteins, and possibly in other proteins whose native conformation is metastable – a hypothesis that shall be studied in future work.*

We have added a paragraph to stress this idea (see below) and have made a few changes throughout the text for the sake of compatibility.

Added paragraph (Discussion, page 10): remaining text from the previous version is black, new text is green.

This suggests a possible causal link between structural vulnerability and affinity to ACE2. The structural features of S allowing the virus to develop its function and to avoid the immune response are in turn those ones particularly unprotected to EFs. *The prefusion state of the S protein of SARS-CoV-2, like that of other class I viral fusion proteins, is metastable [70]. This seems to be important for optimizing or regulating their functions. Thus, function of viral fusion proteins depends on their ability to fold into a less stable but functionally relevant pre-fusion conformation and to limit the kinetic accessibility of the more stable post-fusion conformation. Now, the energy barrier trapping the*

prefusion state was found to be surprisingly low in the case of coronaviruses S proteins [70]. We hypothesise that low to moderate electric fields are enough to modify the energy landscape around the local prefusion minimum, inducing a non-thermal transition to a state between the pre- and the post-fusion conformations. This argument is also supported by our calculation of the free energy profile shown in the Supplementary Fig. 3. As functionally active metastable states are encountered in other viral fusion proteins (e.g., influenza virus hemagglutinin, gp120-gp41 HIV, or Ebola virus GP), we expect the unusual vulnerability to EF to occur for those cases.

----- end of reply to reviewer 3 -----

We have also added an additional text referring to the different lineages of the Delta variant, in which we argue why we expect that it is also at least as vulnerable to EF as the wild type or the other variants studied in our work. This text was also added in the Discussion after summarizing the results on the other variants of concern.

Modified text regarding the Delta variant (Discussion, page 11)

We have shown in this paper that the mutated types Variants of Concern B.1.1.7, B.1.351 and P.1 are at least as strongly damaged by EFs of low to moderate strengths as the wild type RBD. This occurs because their multiple S-protein mutations involve electric charge changes (from neutral to negative in A570D, from neutral to positive in P681H, from negative to positive in E484K and D1118H), while the native prefusion conformation of S is mostly preserved. Other mutants not analysed here, such as the VOC B.1.617 (India), also involve charge substitutions that have been suggested to enhance electrostatic interaction of the RBD with the ACE2 receptor [78]. The concrete case of the sub-variants B.1.617.1 (L452R changes from neutral to positive and E484Q from negative to neutral), B.1.617.2 (L452R changes from neutral to positive and T478K from neutral to positive), and B.1.617.3 (L454R changes from neutral to positive and E484Q from negative to neutral) [79-80] clearly reflect this fact. Therefore, we expect that mutants labelled as variants of interest, under monitoring or that may arise in future, will also be particularly vulnerable to EF. The same should hold for the spike proteins of other viruses.

Together with the inclusion of these two new paragraphs, we have also added two additional references (highlighted on the list). This led to a change of the numbers of the references throughout the paper.

REFERENCES

- [Ref.1] Colman, Peter M., and Michael C. Lawrence. "The structural biology of type I viral membrane fusion." *Nature Reviews Molecular Cell Biology* 4.4 (2003): 309-319.
- [Ref.2] Remington, Jacob M., et al. "Enhanced Sampling Protocol to Elucidate Fusion Peptide Opening of SARS-CoV-2 Spike Protein." *Biophysical Journal* (2021).
- [Ref.3] Kirchdoerfer, Robert N., et al. "Pre-fusion structure of a human coronavirus spike protein." *Nature* 531.7592 (2016): 118-121.
- [Ref.4] Tzarum, Netanel, et al. "Structure and receptor binding of the hemagglutinin from a human H6N1 influenza virus." *Cell host & microbe* 17.3 (2015): 369-376.
- [Ref.5] Takada, A., C. Robison, H. Goto, A. Sanchez, K. G. Murti, M. A. Whitt, and Y. Kawaoka. 1997. A system for functional analysis of Ebola virus glycoprotein. *Proc. Natl. Acad. Sci. USA*94:14764-14769.
- [Ref.6] Ward, Andrew B., and Ian A. Wilson. "The HIV-1 envelope glycoprotein structure: Nailing down a moving target." *Immunological reviews* 275.1 (2017): 21-32.
- [Ref.7] Bosch, Berend Jan, et al. "The coronavirus spike protein is a class I virus fusion protein: structural and functional characterization of the fusion core complex." *Journal of virology* 77.16 (2003): 8801-8811.
- [Ref.8] Baldwin, Andrew J., et al. "Metastability of native proteins and the phenomenon of amyloid formation." *Journal of the American Chemical Society* 133.36 (2011): 14160-14163.
- [Ref.9] Ghosh, Debasish Kumar, and Akash Ranjan. "The metastable states of proteins." *Protein Science* 29.7 (2020): 1559-1568.
- [Ref.10] Baker, David. "Metastable states and folding free energy barriers." *nature structural biology* 5.12 (1998): 1021-1024.

- [Ref.11] Lee, Cheolju, et al. "Regulation of protein function by native metastability." *Proceedings of the National Academy of Sciences* 97.14 (2000): 7727-7731.
- [Ref.12] Khan, Mohammad Sazzad, et al. "Serpin inhibition mechanism: a delicate balance between native metastable state and polymerization." *Journal of amino acids* 2011 (2011).
- [Ref.13] Carr, Chavela M., Charu Chaudhry, and Peter S. Kim. "Influenza hemagglutinin is spring-loaded by a metastable native conformation." *Proceedings of the National Academy of Sciences* 94.26 (1997): 14306-14313.
- [Ref.14] Sohl JL, Jaswal SS, Agard DA (1998) Unfolded conformations of alpha-lytic protease are more stable than its native state. *Nature* 395:817–819.
- [Ref.15] Rao, VV Hemanth Giri, and Shachi Gosavi. "On the folding of a structurally complex protein to its metastable active state." *Proceedings of the National Academy of Sciences* 115.9 (2018): 1998-2003.
- [Ref.16] Chen, Jue, et al. "Structure of the hemagglutinin precursor cleavage site, a determinant of influenza pathogenicity and the origin of the labile conformation." *Cell* 95.3 (1998): 409-417.
- [Ref.17] Sittel, Florian, and Gerhard Stock. "Perspective: Identification of collective variables and metastable states of protein dynamics." *The Journal of chemical physics* 149.15 (2018): 150901.
- [Ref.18] Castin, Jesu E., et al. "pH and Receptor Induced Conformational Changes-Implications Towards S1 Dissociation of SARS-CoV2 Spike Glycoprotein." *bioRxiv* (2020).
- [Ref.19] Galloway, Summer E., et al. "Influenza HA subtypes demonstrate divergent phenotypes for cleavage activation and pH of fusion: implications for host range and adaptation." *PLoS pathogens* 9.2 (2013): e1003151.
- [Ref.20] Benhaim, Mark A., and Kelly K. Lee. "New biophysical approaches reveal the dynamics and mechanics of type I viral fusion machinery and their interplay with membranes." *Viruses* 12.4 (2020): 413.
- [Ref.21] Yin, Hsien-Sheng, et al. "Structure of the parainfluenza virus 5 F protein in its metastable, prefusion conformation." *Nature* 439.7072 (2006): 38-44.
- [Ref.22] Hsieh, Ching-Lin, et al. "Structure-based design of prefusion-stabilized SARS-CoV-2 spikes." *Science* 369.6510 (2020): 1501-1505.
- [Ref.23] McCallum, Matthew, et al. "Structure-guided covalent stabilization of coronavirus spike glycoprotein trimers in the closed conformation." *Nature structural & molecular biology* 27.10 (2020): 942-949.
- [Ref.24] Song, Wenfei, et al. "Cryo-EM structure of the SARS coronavirus spike glycoprotein in complex with its host cell receptor ACE2." *PLoS pathogens* 14.8 (2018): e1007236.
- [Ref.25] Gur, Mert, et al. "Conformational transition of SARS-CoV-2 spike glycoprotein between its closed and open states." *The Journal of Chemical Physics* 153.7 (2020): 075101.
- [Ref.26] Cai, Yongfei, et al. "Distinct conformational states of SARS-CoV-2 spike protein." *Science* 369.6511 (2020): 1586-1592.